# Non-adversarial training of Neural SDEs with signature kernel scores

Zacharia Issa[*1]    Blanka Horvath[2,3,4]    Maud Lemercier[2,4]    Cristopher Salvi[4,5]

## Abstract

Neural SDEs are continuous-time generative models for sequential data. State-of-the-art performance for irregular time series generation has been previously obtained by training these models adversarially as GANs. However, as typical for GAN architectures, training is notoriously unstable, often suffers from mode collapse, and requires specialised techniques such as weight clipping and gradient penalty to mitigate these issues. In this paper, we introduce a novel class of scoring rules on pathspace based on signature kernels and use them as objective for training Neural SDEs non-adversarially. By showing strict properness of such kernel scores and consistency of the corresponding estimators, we provide existence and uniqueness guarantees for the minimiser. With this formulation, evaluating the generator-discriminator pair amounts to solving a system of linear path-dependent PDEs which allows for memory-efficient adjoint-based backpropagation. Moreover, because the proposed kernel scores are well-defined for paths with values in infinite dimensional spaces of functions, our framework can be easily extended to generate spatiotemporal data. Our procedure permits conditioning on a rich variety of market conditions and significantly outperforms alternative ways of training Neural SDEs on a variety of tasks including the simulation of rough volatility models, the conditional probabilistic forecasts of real-world forex pairs where the conditioning variable is an observed past trajectory, and the mesh-free generation of limit order book dynamics.

## 1   Introduction

Stochastic differential equations (SDEs) are a dominant modelling framework in many areas of science and engineering. They naturally extend ordinary differential equations (ODEs) for modelling dynamical systems that evolve under the influence of randomness.

A *neural stochastic differential equation* (Neural SDE) is a continuous-time generative model for irregular time series where the drift and diffusion functions of an SDE are parametrised by neural networks [TR19, JB19, HvdHRM20, LWCD20, KFL+21, MSKF21]. These models have become increasingly popular among financial practitioners for pricing and hedging of derivatives and overall risk management [ASS20, GSVŠ+20, CJB23, HFH+23]. Training a Neural SDE amounts to minimising over model parameters an appropriate notion of distance between the law on pathspace generated by the SDE and the empirical law supported on observed data sample paths.

---

[*]Corresponding author: `zacharia.issa@kcl.ac.uk`

[1]Department of Mathematics, King's College London, London, United Kingdom.

[2]Department of Mathematics, Oxford University, Oxford, United Kingdom.

[3]The Oxford Man Institute, Oxford, United Kingdom.

[4]The Alan Turing Institute, London, United Kingdom.

[5]Department of Mathematics, Imperial College London, London, United Kingdom.

37th Conference on Neural Information Processing Systems (NeurIPS 2023).

Various choices of training mechanisms have been proposed in the literature; state-of-the-art performance has been achieved by training Neural SDEs adversarially as Wasserstein-GANs [KFL+21]. However, as typical for GAN architectures, training is notoriously unstable, often suffers from mode collapse, and requires specialised techniques such as weight clipping and gradient penalty.

In this paper we introduce a novel class of scoring rules based on *signature kernels*, a class of characteristic kernels on paths [LSD+21, CFC+21, SLL+21, LSC+21, CLS23, HLL+23], and use them as objective for training Neural SDEs non-adversarially. We provide existence and uniqueness guarantees for the minimiser by showing strict properness of the signature kernel scores and consistency of the corresponding estimators.

With this training formulation, the generator-discriminator pair becomes entirely mesh-free and can be evaluated by solving a system of linear path-dependent PDEs which allows for memory-efficient adjoint-based backpropagation. In addition, because the proposed kernel scores are well-defined for classes of paths with values in infinite dimensional spaces of functions, our framework can be easily extended to the generation of spatiotemporal signals.

We demonstrate how our procedure is more stable and outperforms alternative ways of training Neural SDEs on a variety of tasks from quantitative finance including the simulation of rough volatility models, the conditional probabilistic forecasts of real-world forex pairs where the conditioning variable is an observed past trajectory, and the mesh-free generation of limit order book dynamics.

## 2  Related work

Prior to our work, two main approaches have been proposed to fit a Neural SDE as a time series generative model, differing in their choice of divergence to compare laws on pathspace.

The SDE-GAN model introduced in [KFL+21] uses the 1-Wasserstein distance to train a Neural SDE as a Wasserstein-GAN [ACB17]. Namely, the "witness functions" of the 1-Wasserstein distance are parameterised by neural controlled differential equations [KMFL20, MSK+20] and the generator-discriminator pair is trained adversarially. SDE-GANs are relatively unstable to train mainly because they require a Lipschitz discriminator. Several techniques such as weight clipping and gradient penalty have been introduced to enforce the Lipschitz constraint and partially mitigate the instability issue [Kid22]. SDE-GANs are also sensitive to other hyperparameters, such as the choice of optimisers, their learning rate and momentum, where small changes can yield erratic behavior.

The latent SDE model [LWCD20] trains a Neural SDE with respect to the KL divergence using the principles of variational inference for SDEs [Opp19]. This approach consists in maximising an objective that includes the KL divergence between the laws produced by the original SDE (the prior) and an auxiliary SDE (the approximate posterior). The two SDEs have the same diffusion term but different initial conditions and drifts, and a standard formula for their KL divergence exists. After training, the learned prior can be used to generate new sample paths. Latent SDEs can be interpreted as variational autoencoders, and generally yield worse performance than SDE-GANs, which are more challenging to train, but offer greater model capacity.

Besides Neural SDEs, other time series generative models have been proposed, including discrete-time models such as [YJVdS19] and [NSW+20][2] which are trained adversarially, continuous-time flow processes [DCB+20] and score-based diffusion models for audio generation [CZZ+20, KPH+20].

The class of score-based generative models (SGMs) seeks to map a data distribution into a known prior distribution via an SDE [SSDK+20, VKK21]. During training, the (Stein) score [LLJ16] of the SDE marginals is estimated and then used to construct a reverse-time SDE. By sampling data from the prior and solving the reverse-time SDE, one can generate samples that follow the original data distribution. We note that our techniques for generative modelling via scoring rules, although similar in terminology, are fundamentally different, as we train Neural SDEs with respect to a loss function that directly consumes the law on pathspace generated by the SDE.

Scoring rules [GR07] have been used to define training objectives for generative networks [BMN16, GSvdB+20] which have been shown to be easier to optimize compared to GANs [PADD21, PD22]. Closer to our work is [PADD21] which constructs statistical scores for discrete (spatio-)temporal

---

[2]In [NSW+20] the discriminator is formulated in continuous-time based on a different parametrisation to approximate the 1-Wasserstein distance, also later used in [NSSV+21].

signals. However, their strict properness is ensured under Markov-type assumptions and their continuous-(space-)time limit has not been studied. A key aspect of our work is to develop consistent and effective scoring rules for generative modelling in the continuous-time setting. While [BO21] has also introduced scoring rules for continuous-time processes, our emphasis lies in constructing so-called kernel scores specifically for training Neural SDE and Neural SPDE generative models.

The Neural SPDE model introduced in [SLG22] parametrises the solution operator of stochastic partial differential equations (SPDEs), which extend SDEs for modelling signals that vary both in space and in time. So far, Neural SPDEs have been trained in a supervised fashion by minimizing the pathwise $L^2$ norm between pairs of spatiotemporal signals. While this approach has proven effective in learning fast surrogate SPDE solvers, it is not well-suited for generative modeling where the goal is to approximate probability measures supported on spatiotemporal functions. In this work, we propose a new training objective for Neural SPDEs to improve their generative modeling capabilities.

## 3 Training Neural SDEs with signature kernel scores

### 3.1 Background

We take $(\Omega, \mathcal{F}, \mathbb{P})$ as the underlying probability space. Let $T > 0$ and $d_x \in \mathbb{N}$. Denote by $\mathcal{X}$ be the space of continuous paths of bounded variation from $[0, T]$ to $\mathbb{R}^{d_x}$ with one monotone coordinate[3]. For any random variable $X$ with values on $\mathcal{X}$, we denote by $\mathbb{P}_X := \mathbb{P} \circ X^{-1}$ its law.

The *signature map* $S : \mathcal{X} \to \mathcal{T}$ is defined for any path $x \in \mathcal{X}$ as the infinite collection $S(x) = \left(1, S^1(x), S^2(x), ...\right)$ of iterated Riemann-Stieltjes integrals

$$S^k(x) := \int_{0 < t_1 < ... < t_k < T} dx_{t_1} \otimes dx_{t_2} \otimes ... \otimes dx_{t_k}, \quad k \in \mathbb{N},$$

where $\otimes$ is the standard tensor product of vector spaces and $\mathcal{T} := \mathbb{R} \oplus \mathbb{R}^{d_x} \oplus (\mathbb{R}^{d_x})^{\otimes 2} \oplus ...$

Any inner product $\langle \cdot, \cdot \rangle_1$ on $\mathbb{R}^{d_x}$ yields a canonical Hilbert-Schmidt inner product $\langle \cdot, \cdot \rangle_k$ on $(\mathbb{R}^{d_x})^{\otimes k}$ for any $k \in \mathbb{N}$, which in turn yields, by linearity, a family of inner products $\langle \cdot, \cdot \rangle_{\mathcal{T}}$ on $\mathcal{T}$. We refer the reader to [CLX21] for an in-depth analysis of different choices. By a slight abuse of notation, we use the same symbol to denote the Hilbert space obtained by completing $\mathcal{T}$ with respect to $\langle \cdot, \cdot \rangle_{\mathcal{T}}$.

### 3.2 Neural SDEs

Let $W : [0, T] \to \mathbb{R}^{d_w}$ be a $d_w$-dimensional Brownian motion and $a \sim \mathcal{N}(0, I_{d_a})$ be sampled from $d_a$-dimensional standard normal. The values $d_w, d_a \in \mathbb{N}$ are hyperparameters describing the size of the noise. A Neural SDE is a model of the form

$$Y_0 = \xi_\theta(a), \quad dY_t = \mu_\theta(t, Y_t)dt + \sigma_\theta(t, Y_t) \circ dW_t, \quad X_t^\theta = A_\theta Y_t + b_\theta \tag{1}$$

for $t \in [0, T]$, with $Y : [0, T] \to \mathbb{R}^{d_y}$ the strong solution, if it exists, to the Stratonovich SDE, where

$$\xi_\theta : \mathbb{R}^{d_a} \to \mathbb{R}^{d_y}, \quad \mu_\theta : [0, T] \times \mathbb{R}^{d_y} \to \mathbb{R}^{d_y}, \quad \sigma_\theta : [0, T] \times \mathbb{R}^{d_y} \to \mathbb{R}^{d_y \times d_w}$$

are suitably regular neural networks, and $A_\theta \in \mathbb{R}^{d_x \times d_y}, b_\theta \in \mathbb{R}^{d_x}$. The dimension $d_y \in \mathbb{N}$ is a hyperparameter describing the size of the hidden state. If $\mu_\theta, \sigma_\theta$ are Lipschitz and $\mathbb{E}_a[\xi_\theta(a)^2] < \infty$ then the solution $Y$ exists and is unique.

Given a target $\mathcal{X}$-valued random variable $X^{\text{true}}$ with law $\mathbb{P}_{X^{\text{true}}}$, the goal is to train a Neural SDE so that the generated law $\mathbb{P}_{X^\theta}$ is as close as possible to $\mathbb{P}_{X^{\text{true}}}$, for some appropriate notion of closeness.

### 3.3 Signature kernels scores

*Scoring rules* are a well-established class of functionals to represent the penalty assigned to a distribution given an observed outcome, thereby providing a way to assess the quality of a probabilistic forecast. Scoring rules have been applied to a wide range of areas including econometrics [MS13],

---

[3]This is a technical assumption needed to ensure charactersticness of the signature kernel. See Proposition (3.1). The monotone coordinate is usually taken to be time.

weather forecasting [GR05], and generative modelling [PADD21]. How to effectively select a scoring rule is a challenging and somewhat task-dependent problem, particularly when the data is sequential. Scoring rules based on kernels offer the advantages of working on unstructured and infinite dimensional data without some of the concomitant drawbacks, such as the absence of densities. Next, we introduce a class of scoring rules on paths based on signature kernels to measure closeness between path-valued random variables. These will be used in the next section to train Neural SDEs.

The *signature kernel* $k_{\mathrm{sig}} : \mathcal{X} \times \mathcal{X} \to \mathbb{R}$ is a symmetric positive semidefinite function defined for any pair of paths $x, y \in \mathcal{X}$ as $k_{\mathrm{sig}}(x, y) := \langle S(x), S(y) \rangle_{\mathcal{T}}$. In [SCF$^+$21] the authors provided a kernel trick proving that the signature kernel satisfies

$$k_{\mathrm{sig}}(x,y) = f(T,T) \quad \text{where} \quad f(s,t) = 1 + \int_0^s \int_0^t f(u,v) \langle dx_u, dy_v \rangle_1, \tag{2}$$

which reduces to a linear hyperbolic PDE in the when the paths $x, y$ are almost-everywhere differentiable. Several finite difference schemes are available for numerically evaluating solutions to Equation (2), see [SCF$^+$21, Section 3.1] for details.

We denote by $\mathcal{H}$ the unique reproducing kernel Hilbert space (RKHS) of $k_{\mathrm{sig}}$. From now on we endow $\mathcal{X}$ with a topology with the respect to which the signature is continuous; see [CT22] for various choices of such topologies. Denote by $\mathcal{P}(\mathcal{X})$ the set of Borel probability measures on $\mathcal{X}$.

**Proposition 3.1.** *The signature kernel is characteristic for every compact set $\mathcal{K} \subset \mathcal{X}$, i.e. the map $\mathbb{P} \mapsto \int k_{sig}(x, \cdot) \, \mathbb{P}(dx)$ from $\mathcal{P}(\mathcal{K})$ to $\mathcal{H}$ is injective.*

**Remark 3.2.** The proof of this statement is classical and is a simple consequence of the universal approximation property of the signature [KBPA$^+$19, Proposition A.6] and the equivalence between universality of the feature map and characteristicness of the corresponding kernel [SGS18, Theorem 6]. In particular, Proposition (3.1) implies that the signature kernel is cc-universal, i.e. for every compact subset $\mathcal{K} \subset \mathcal{X}$, the linear span of the set of path functionals $\{k_{\mathrm{sig}}(x, \cdot) : x \in \mathcal{K}\}$ is dense in $C(\mathcal{K})$ in the the topology of uniform convergence.

We define the *signature kernel score* $\phi_{\mathrm{sig}} : \mathcal{P}(\mathcal{X}) \times \mathcal{X} \to \mathbb{R}$ for any $\mathbb{P} \in \mathcal{P}(\mathcal{X})$ and $y \in \mathcal{X}$ as

$$\phi_{\mathrm{sig}}(\mathbb{P}, y) := \mathbb{E}_{x, x' \sim \mathbb{P}}[k_{\mathrm{sig}}(x, x')] - 2\mathbb{E}_{x \sim \mathbb{P}}[k_{\mathrm{sig}}(x, y)].$$

A highly desirable property to require from a score is its *strict properness*, consisting in assigning the lowest expected score when the proposed prediction is realised by the true probability distribution.

**Proposition 3.3.** *For any compact $\mathcal{K} \subset \mathcal{X}$, $\phi_{sig}$ is a strictly proper kernel score relative to $\mathcal{P}(\mathcal{K})$, i.e. $\mathbb{E}_{y \sim \mathbb{Q}}[\phi_{sig}(\mathbb{Q}, y)] \leq \mathbb{E}_{y \sim \mathbb{Q}}[\phi_{sig}(\mathbb{P}, y)]$ for all $\mathbb{P}, \mathbb{Q} \in \mathcal{P}(\mathcal{K})$, with equality if and only if $\mathbb{P} = \mathbb{Q}$.*

The proof of this statement can be found in the appendix and follows from [GR07, Theorem 4] and Proposition 3.1. We note that the signature kernel score induces a divergence on $\mathcal{P}(\mathcal{X})$ known as the signature kernel *maximum mean discrepancy* (MMD), defined for any $\mathbb{P}, \mathbb{Q} \in \mathcal{P}(\mathcal{X})$ as

$$\mathcal{D}_{k_{\mathrm{sig}}}(\mathbb{P}, \mathbb{Q})^2 = \mathbb{E}_{y \sim \mathbb{Q}}[\phi_{\mathrm{sig}}(\mathbb{P}, y)] + \mathbb{E}_{y, y' \sim \mathbb{Q}}[k_{\mathrm{sig}}(y, y')]. \tag{3}$$

The following result provides a consistent and unbiased estimator for evaluating the signature kernel score from observed sample paths. The proof can be found in the appendix and follows from standard results for the associated MMD [GBR$^+$12, Lemma 6].

**Proposition 3.4.** *Let $\mathbb{P} \in \mathcal{P}(\mathcal{X})$ and $y \in \mathcal{X}$. Given $m$ sample paths $\{x^i\}_{i=1}^m \sim \mathbb{P}$, the following is a consistent and unbiased estimator of $\phi_{sig}$*

$$\widehat{\phi}_{\mathrm{sig}}(\mathbb{P}, y) = \frac{1}{m(m-1)} \sum_{j \neq i} k_{\mathrm{sig}}(x^i, x^j) - \frac{2}{m} \sum_i k_{\mathrm{sig}}(x^i, y). \tag{4}$$

### 3.4 Non-adversarial training of Neural SDEs via signature kernel scores

We now have all the elements to outline the procedure we propose to train the Neural SDE model (1) non-adversarially using signature kernel scores introduced in the previous section.

**Unconditional setting**  We are given a target $\mathcal{X}$-valued random variable $X^{\text{true}}$ with law $\mathbb{P}_{X^{\text{true}}}$. Recall the notation $\mathbb{P}_{X^\theta}$ for the law generated by the SDE (1). The training objective is given by

$$\min_\theta \mathcal{L}(\theta) \quad \text{where} \quad \mathcal{L}(\theta) = \mathbb{E}_{y \sim \mathbb{P}_{X^{\text{true}}}} [\phi_{\text{sig}}(\mathbb{P}_{X^\theta}, y)]. \tag{5}$$

Note that training with respect to $\mathcal{D}_{k_{\text{sig}}}$ is an equivalent optimisation as the second expectation in equation (3) is constant with respect to $\theta$. This means that in the unconditional setting our model corresponds to a continuous time generative network of [LSZ15].

Combining equations (1), (2), (4) and (5) the generator-discriminator pair can be evaluated by solving a system of linear PDEs depending on sample paths from the Neural SDE; in summary:

**Generator:** $X^\theta \sim \text{SDESolve}(\theta)$    **Discriminator:** $\mathcal{L}(\theta) \approx \text{PDESolve}\left(X^\theta, X^{\text{true}}\right).$    (6)

**Remark 3.5.** The generation of sample paths from $X^\theta$ from the SDE solver and the evaluation of the objective $\mathcal{L}$ via the PDE solver can in principle be performed concurrently, although, in our implementation we evaluate the full model (6) in a sequential manner.

**Conditional setting**  It is straightforward to extend our framework to the conditional setting where $\mathbb{Q}$ is some distribution we wish to condition on, and $\mathbb{P}_{X^{\text{true}}}(\cdot|x)$ is a target conditional distribution with $x \sim \mathbb{Q}$. By feeding the observed sample $x$ as an additional variable to all neural networks of the Neural SDE (1), the generated strong solution provides a parametric conditional law $\mathbb{P}_{X^\theta}(\cdot|x)$, and the model can be trained according to the modified objective

$$\min_\theta \mathcal{L}'(\theta) \quad \text{where} \quad \mathcal{L}'(\theta) = \min_\theta \mathbb{E}_{x \sim \mathbb{Q}} \mathbb{E}_{y \sim \mathbb{P}_{X^{\text{true}}}(\cdot|x)} [\phi_{\text{sig}}(\mathbb{P}_{X^\theta}(\cdot|x), y)]. \tag{7}$$

Because $\phi_{\text{sig}}$ is strictly proper, the solution to (7) is $\mathbb{P}_{X^\theta}(\cdot|x) = \mathbb{P}_{X^{\text{true}}}(\cdot|x)$ $\mathbb{Q}$-almost everywhere. With data sampled as $\{(x^i, y^i)\}_{i=1}^n$ where $x^i \sim \mathbb{Q}$ and $y^i \sim \mathbb{P}_{X^{\text{true}}}(\cdot|x^i)$ we can replace eq. (7) by

$$\min_\theta \frac{1}{n} \sum_{i=1}^n \phi_{\text{sig}}(\mathbb{P}_{X^\theta}(\cdot|x^i), y^i), \tag{8}$$

We note that in our experiments we focus on the specific case where the conditioning variable $x$ is a path in $\mathcal{X}$ corresponding to the observed past trajectory of some financial assets (see Figure 2).

### 3.5  Additional details

**Interpolation**  Samples from $X^{\text{true}}$ are observed on a discrete, possibly irregular, time grid while samples from $X^\theta$ are generated from (1) by means of an SDE solver of choice (see [Kid22, Section 5.1] for details). Interpolating in time between observations produces a discrete measure on path space, the ones desired to be modelled. The interpolation choice is usually unimportant and simple linear interpolation is often sufficient. See [MKYL22] for other choices of interpolation.

**Backpropagation**  Training a Neural SDE usually means backpropagating through the SDE solver. Three main ways of differentiating through an SDE have been studied in the literature: 1) *Discretise-then-optimise* backpropagates through the internal operations of the SDE solver. This option is memory inefficient, but will produce accurate and fast gradient estimates. 2) *Optimise-then-discretise* derives a backwards-in-time SDE, which is then solved numerically. This option is memory efficient, but gradient estimates are prone to numerical errors and generally slow to compute. We note that unlike the case of Neural ODEs, giving a precise meaning to the backward SDE falls outside the usual framework of diffusions. However, *rough path theory* [Lyo98, FLMS23] provides an elegant remedy by allowing solutions to forward and backward SDEs to be defined pathwise, similarly to the case of ODEs; see [Kid22, Appendix C.3.3] for a precise statement. 3) *Reversible solvers* are memory efficient and accurate, but generally slow. Here we do not advocate for any particular choice as all of the above backpropagation options are compatible with our pipeline.

Similarly, because the signature kernel score can be evaluated by solving a system of PDEs, backpropagation can be carried out by differentiating through the PDE solver analogously to the discretise-then-optimise option for SDEs. We note that [LSC$^+$21] showed that directional derivatives of signature kernels solve a system of adjoint-PDEs, which can be leveraged to backpropagate through the discriminator using an optimise-then-discretise approach. We used this approach in our experiments.

**Itô vs Stratonovich**    Stratonovich SDEs are slightly more efficient to backpropagate through using an optimise-then-discretise approach. In the case of Itô SDEs, the backward equation is derived by applying the Itô-Stratonovich correction term to convert it into a Stratonovich SDE, deriving the corresponding backward equation through rough path theoretical arguments, and then converting it back to an Itô SDE by applying a second Stratonovich-Itô correction.

**Paths with values in infinite dimensional spaces**    While we have defined the signature kernel for paths of bounded variation with values in $\mathbb{R}^{d_x}$, the kernel is still well-defined when $\mathbb{R}^{d_x}$ is replaced with a generic Hilbert space $V$. Remarkably, even when $V$ is infinite dimensional, the evaluation of the kernel can be carried out, as Equation (2) only depends on pairwise inner products between the values of the input paths. In particular, the kernel can be evaluated on paths taking their values in functional spaces, which has far-reaching consequences in practice. For example, this gives the flexibility to map the values of finite dimensional input paths into a possibly infinite dimensional feature space, such as the reproducing kernel Hilbert space of a kernel $\kappa$ on $\mathbb{R}^{d_x}$, that is, $V = \mathcal{H}_\kappa$. This also provides a natural kernel for spatiotemporal signals, such as paths taking their values in $V = L^2(D)$, the space of square-integrable functions on a compact domain $D \subset \mathbb{R}^d$. For practical applications, the inner product in Equation (2) can be approximated using discrete observations of the input signals on a mesh of the spatial domain $D$. The inner product in $L^2(D)$ can be replaced with more general kernels as those introduced in [WD22]. While it has become common practice to use signature kernels on the RKHS-lifts of Euclidean-valued paths, the ability to define and compute signature kernels on spatiotemporal signals has been, to our knowledge, overlooked in the literature.

## 4    Experiments

We perform experiments across five datasets. First is a univariate synthetic example, the benchmark Black-Scholes model, which permits to readily verify the quality of simulated outputs. The second synthetic example is a state of-the-art univariate stochastic volatility model, called rough Bergomi model. The rough Bergomi model realistically captures many relevant properties of options data, but due to its rough (and hence non-Markovian) nature it is well-known to be difficult to simulate. The third is a multidimensional example with foreign exchange (forex, or FX) currency pairs, which was chosen not only because of the relevance and capitalisation of FX markets but also due to its well-known intricate complexity. Fourth is a univariate example, where we demonstrate the method's ability to condition on relevant variables, given by paths. Finally we present a spatiotemporal generative example, where we seek to simulate the dynamics of the NASDAQ limit order book.

For the unconditional examples, we compare against the SDE-GAN from [KFL+21] and against the same pipeline as the one we proposed, but using an approximation $\phi_{\text{sig}}^N$ of the signature kernel score $\phi_{\text{sig}}$ obtained by truncating signatures at some level $N \in \mathbb{N}$. We evaluate each training instance with a variety of metrics. The first is the Kolmogorov-Smirnov (KS) test, which is a nonparameteric two-sample test between two empirical probability distributions on $\mathbb{R}$, see [Smi39] for more details. We apply the KS test on the marginals between a batch of generated paths against an unseen batch from the real data distribution. We repeated this test 5000 times at the 5% significance level and reported the average KS score along with the average Type I error. Each training instance was kept to a maximum of 2 hours for the synthetic examples, and 4 hours for the real data example. Finally, as mentioned at the end of Section 3.5, when training with respect to $\phi_{\text{sig}}$ we mapped path state values into $(\mathcal{H}, \kappa)$ where $\kappa$ denotes the RBF kernel on $\mathbb{R}^d$. Additional details on hyperparameter selection, learning rates, optimisers and further evaluation metrics can be found in the Appendix.

### 4.1    Geometric Brownian motion

As a toy example, we seek to learn a *geometric Brownian motion* (gBm) of the form

$$dy_t = \mu y_t dt + \sigma y_t dW_t, \qquad y_0 = 1, \tag{9}$$

We chose $\mu = 0, \sigma = 0.2$ and generated time-augmented paths of length 64 over the grid $\Delta = \{0, 1, \ldots, 63\}$ with $dt = 0.01$. Thus our dataset is given by time-augmented paths $y : [0, 63] \to \mathbb{R}^2$ embedded in path space via linear interpolation. For all three discriminators, the training and test set were both comprised of 32768 paths and the batch size was chosen to be $N = 128$. We trained the SDE-GAN for 5000 steps, $\phi_{\text{sig}}$ for 4000 and $\phi_{\text{sig}}^N$ for 10000 steps. Table 1 gives the KS scores along

each of the specified marginals, along with the percentage Type I error. Here the generator trained with $\phi_{\text{sig}}$ performs the best, achieving a Type I error at the assumed confidence level.

| Model | $t = 6$ | $t = 19$ | $t = 32$ | $t = 44$ | $t = 57$ |
|---|---|---|---|---|---|
| SDE-GAN | $0.1641, 41.1\%$ | $\mathbf{0.1094, 5.2\%}$ | $0.1421, 24.2\%$ | $0.1104, 5.9\%$ | $0.1427, 26.2\%$ |
| $\phi_{\text{sig}}^N$ ($N = 3$) | $0.1298, 15.4\%$ | $0.1277, 16.1\%$ | $0.1536, 37.4\%$ | $0.2101, 78.8\%$ | $0.2416, 92.3\%$ |
| $\phi_{\text{sig}}$ (ours) | $\mathbf{0.1071, 5.0\%}$ | $0.1084, 6.0\%$ | $\mathbf{0.1086, 5.9\%}$ | $\mathbf{0.1089, 5.8\%}$ | $\mathbf{0.1075, 5.5\%}$ |

Table 1: KS test average scores and Type I errors on marginals on gBm.

## 4.2 Rough Bergomi volatility model

It is well-known that the benchmark model (9) oversimplifies market reality. More complex models, *(rough) stochastic volatility* (SV) were introduced in the past decades, that are able to capture relevant properties of market data are used by financial practitioners to price and hedge derivatives. Prominent examples of stochastic volatility mdoels include the Heston and SABR models [HKLW02, HLW15, Hes93]. State-of-the-art models in this context have been introduced in [GJR18]. They display a stochastic volatility with *rough* sample paths. Most notable among these for pricing and hedging is the *rough Bergomi* (rBergomi) model [BFG16] which is of the form

$$dy_t = -\frac{1}{2}V_t dt + \sqrt{V_t} dW_t \quad \text{where} \quad d\xi_t^u = \xi_t^u \eta \sqrt{2\alpha + 1}(u - t)^\alpha dB_t, \qquad (10)$$

and where $\xi_t^u$ is the instantaneous forward variance for time $u$ at time $t$, with $\xi_t^t = V_t$, and $\alpha = H - 1/2$ where $H$ is the Hurst exponent. The parameter set is given by $(\eta, \rho, H)$ with initial conditions $X_0 = x$ and $\xi_t^u = \xi_0$. It has been a well-known headache for modellers that—despite their many modelling advantages—rough volatility models (such as (10)) are slow to to simulate with traditional methods. We demonstrate how our method can be used to capture the dynamics of the rough Bergomi model (10), and in passing we also note that our method provides a significant simulation speedup for (10) compared to previously available simulation methods.

| Model | $t = 6$ | $t = 19$ | $t = 32$ | $t = 44$ | $t = 57$ |
|---|---|---|---|---|---|
| SDE-GAN | $0.1929, 68.3\%$ | $0.2244, 86.2\%$ | $0.2273, 87.0\%$ | $0.2205, 83.4\%$ | $0.1949, 68.7\%$ |
| $\phi_{\text{sig}}^N$ ($N = 5$) | $0.1126, 8.1\%$ | $0.1172, 10.1\%$ | $0.1146, 8.2\%$ | $0.1153, 8.5\%$ | $0.1134, 7.0\%$ |
| $\phi_{\text{sig}}$ (ours) | $\mathbf{0.1086, 5.4\%}$ | $\mathbf{0.1129, 5.9\%}$ | $\mathbf{0.1118, 5.2\%}$ | $\mathbf{0.1127, 6.2\%}$ | $\mathbf{0.1159, 6.9\%}$ |

Table 2: KS test average scores and Type I errors on marginals on rBergomi model

To do so, we simulate paths of length 64 over the time window to $[0, 2]$, and specify $dt = 1/32$. Thus paths are of length 64. The parameters are $(\xi_0, \eta, \rho, H) = (0.04, 1.5, -0.7, 0.2)$ and set $d = 1$. Paths are again time-augmented. The hyperparameters for training are the same as in the previous section. The results on the marginal distributions are summarized in Table 2. We see that that training with respect to $\phi_{\text{sig}}$ vastly outperforms the other two discriminators.

## 4.3 Foreign exchange currency pairs

We consider an example where samples from the data measure $\mathbb{P}_{X^{\text{true}}}$ are time-augmented paths $y : [0, T] \to \mathbb{R}^3$ corresponding to hourly market close prices of the currency pairs EUR/USD and USD/JPY[4]. To deal with irregular sampling, we linearly interpolate each sample $y$ over a fixed grid $\Delta = \{t_0, t_1, \ldots, t_{63}\}$. Training hyperparameters were kept the same as per the rBergomi example: paths are comprised of 64 observations, the batch size was taken to be $N = 128$, and the number of training epochs was taken to be 10000 for the SDE-GAN, 4000 for $\phi_{\text{sig}}$ and 15000 for $\phi_{\text{sig}}^N$. KS scores for each of the marginals are given in Table 3 and 4. We note that only the generator trained with $\phi_{\text{sig}}$ is able to achieve strong performance on nearly all marginals.

---

[4]Data is obtained from `https://www.dukascopy.com/swiss/english/home/`

| Model | $t=6$ | $t=19$ | $t=32$ | $t=44$ | $t=57$ |
|---|---|---|---|---|---|
| SDE-GAN | $0.1889, 62.9\%$ | $0.2760, 98.2\%$ | $0.3324, 99.9\%$ | $0.3781, 100.0\%$ | $0.4209, 100.0\%$ |
| $\phi_{\text{sig}}^N$ ($N=5$) | **0.1098, 4.2%** | $0.1279, 12.0\%$ | $0.1399, 18.7\%$ | $0.1507, 28.1\%$ | $0.1608, 37.5\%$ |
| $\phi_{\text{sig}}$ (ours) | $0.1270, 12.8\%$ | **0.1085, 5.2%** | **0.1060, 4.3%** | **0.1065, 5.1%** | **0.1049, 4.0%** |

Table 3: KS test average scores on marginals (EUR/USD)

| Model | $t=6$ | $t=19$ | $t=32$ | $t=44$ | $t=57$ |
|---|---|---|---|---|---|
| SDE-GAN | $0.1404, 20.5\%$ | $0.1665, 44.2\%$ | $0.1771, 56.4\%$ | $0.1855, 63.8\%$ | $0.1948, 70.3\%$ |
| $\phi_{\text{sig}}^N$ ($N=5$) | $0.1666, 43.8\%$ | $0.1877, 72.4\%$ | $0.2008, 84.7\%$ | $0.2154, 93.2\%$ | $0.2311, 98.3\%$ |
| $\phi_{\text{sig}}$ (ours) | **0.1189, 9.2%** | **0.1121, 5.8%** | **0.1069, 4.9%** | **0.1075, 3.8%** | **0.1051, 3.3%** |

Table 4: KS test average scores on marginals (USD/JPY).

We also present a histogram of sample correlations between generated EUR/USD and USD/JPY paths for each of the three discriminators alongside those from the data distribution. From Figure 1 it appears that only the Neural SDE trained with $\phi_{\text{sig}}$ correctly identifies the negative correlative structure between the two pairs. This is likely due to the fact that these dependencies are encoded in higher order terms of the signature that the truncated method does not capture.

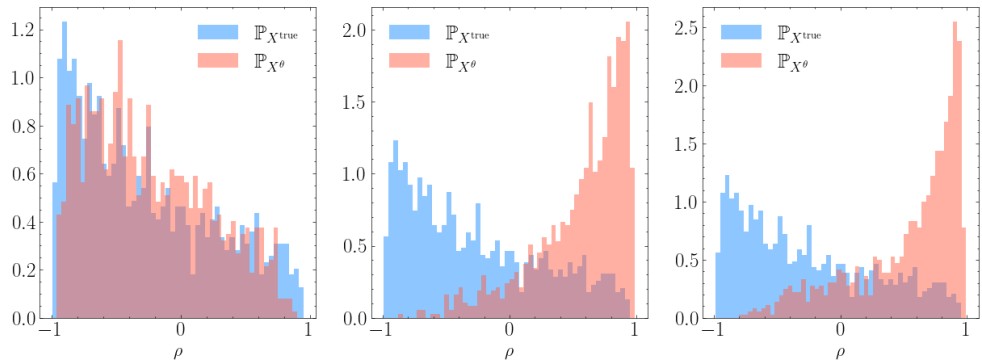

Figure 1: Histogram of correlation coefficients between EURUSD and USDJPY pairs, 1024 samples.

We now consider a conditional generation problem, where the conditioning variables are time-augmented paths $\mathbb{Q} \sim x : [t_0 - dt, t_0] \to \mathbb{R}^2$ representing the trajectory of prior $dt = 32$ observations of EUR/USD 15-minute close prices, and the target distribution is $X^{\text{true}} : [t_0, t_0 + dt'] \to \mathbb{R}^2$ representing the following $dt' = 16$ observations. Given batched samples $\{x^i, y^i\}_{i=1}^N$, where $x^i \sim \mathbb{Q}$ and $y^i \sim \mathbb{P}_{X^{\text{true}}}(\cdot | x^i)$, we train our generator according to equation (7). We encoded the conditioning paths via the truncated (log)signatures of order 5, and fed these values into each of the neural networks of the Neural SDE. In Figure 2, it is evident that the conditional generator exhibits the capability to produce conditional distributions that frequently encompass the observed path. Furthermore, it is noteworthy that these generated distributions capture certain distinctive characteristics of financial markets, such as martingality, mean reversion, or leverage effects when applicable.

## 4.4 Simulation of limit order books

Here, we consider the task of simulating the dynamics of a limit order book (LOB), that is, an electronic record of all the outstanding orders for a financial asset, representing its supply and demand over time. Simulating LOB dynamics is an important challenge in quantitative finance and several synthetic market generators have been proposed [LWL+20],[VBP+20],[SCC21],[CPC+21],[CMVB22]. An order $o = (t_o, x_o, v_o)$ submitted at time $t_o$ with price $x_o$ and size $v_o > 0$ (resp., $v_o < 0$) is a commitment to sell (resp., buy) up to $|v_o|$ units of the traded asset at a price no less (resp., no greater)

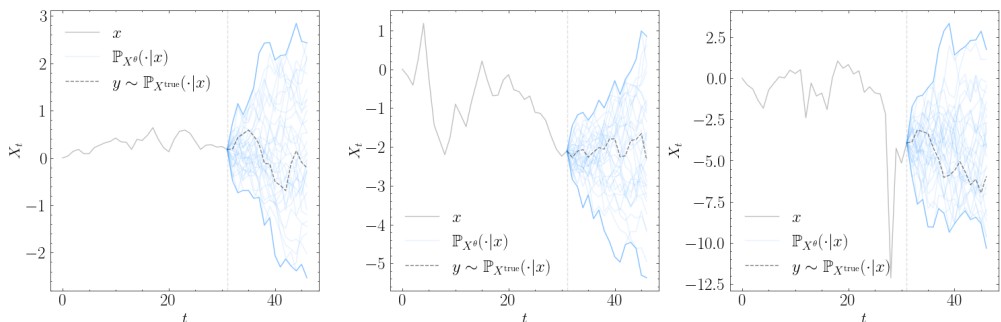

Figure 2: Given a conditioning path $x \sim \mathbb{Q}$, the generator provides (in blue) the conditional distribution $\mathbb{P}_{X^\theta}(\cdot|x)$. The dotted line gives the true path $y \sim \mathbb{P}_{X^{\text{true}}}(\cdot|x)$.

than $x_o$. Various events are tracked (e.g. new orders, executions, and cancellations) and the LOB $\mathcal{B}(t)$ is the set of all active orders in a market at time $t$. While prior work typically fit a generator that produces the next event, and run it iteratively to generate a sequence of events, we propose to model directly the spatiotemporal process $Y_t(x) = \sum_{o \in \mathcal{B}(t):x_o=x} v_o$. To generate LOB trajectories, we use the Neural SPDE model and train it by minimising expected spatiotemporal kernel scores constructed by composing the signature kernel $k_{\text{sig}}$ with 3 different SE-T type kernels introduced in [WD22], namely the ID, SQR and CEXP kernels. We fit our model on real LOB data from the NASDAQ public exchange [NMK$^+$18] which consists of about $4M$ timestamped events with $L = 10$ price levels. We split this LOB trace into sub-traces of size $T = 30$ to construct our dataset. On Figure 3 report the average KS scores for each of the $L \times T$ marginals, using the 3 different kernel scores.

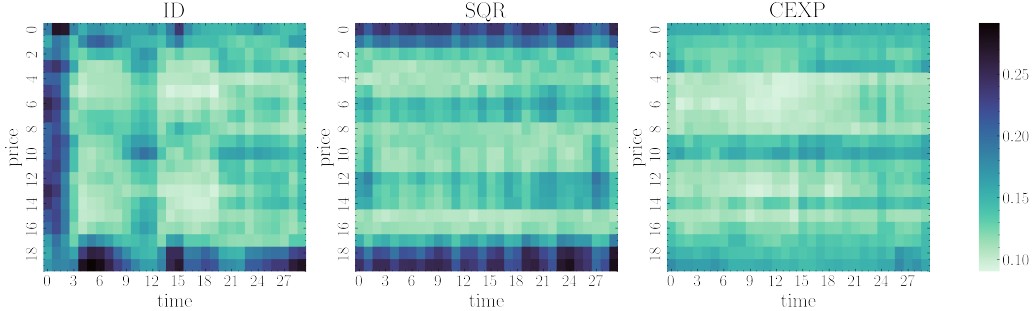

Figure 3: KS test average scores for each spatiotemporal marginal, 100 runs, NASDAQ data.

## 5    Conclusion and future work

This work showcases the utilization of Neural SDEs as a generative model, highlighting their advantages over competitor models in terms of simplicity and stability, particularly via non-adversarial training. Additionally, we show how Neural SDEs exhibit the ability to be conditioned on diverse and intricate data structures, surpassing the capabilities of existing competitor works. We have achieved this by introducing the signature kernel score on paths and by showing their applicability to our setting (by proving strict properness). Performance of our methods are given computational time and memory is competitive with state-of-the-art methods. Moreover, we have shown that this approach extends to the generation of spatiotemporal signals, which has multiple applications in finance including limit order data generation. Further extensions of this work may include extending its generality to include jump processes in the driving noise of the approximator process (Neural SDEs) used. On the theoretical level extensions may include the validity of results to paths with lower regularity than currently considered. Although sample paths from a Stratonovich SDE are not of bounded variation almost surely, sample paths generated by an SDE solver, once interpolated, are piecewise linear, and hence of bounded variation. A similar point can be made about compactness of the support of the measures. It is possible to ensure characteristicness of the signature kernel on non-compact sets of less regular paths using limiting arguments and changing the underlying

topology on pathspace. Further extensions for practical applications can (and should) include the inclusion of more varied evaluation metrics and processes. Notably, in a later step, the generated data should be tested by assessing whether existing risk management frameworks and investment engines can be improved when data used for backtesting is augmented with synthetic samples provided by our methods. Furthermore, the spatiotemporal results can be extended to more complex structures, including being used for the synthetic generation of implied volatility surface dynamics, which has been a notoriously difficult modelling problem in past decades.

## Acknowledgements

ML was supported by the EPSRC grant EP/S026347/1. ZI was supported by EPSRC grant EP/R513064/1.

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

## A  Signature Kernel Scores

**Proof of Proposition 3.3**

*Proof (Appendix).* The general result was first shown in [GR07]. We first show that $\phi_{\text{sig}}$ is proper. By Proposition 3.1 the signature kernel is positive definite and characteristic on $\mathcal{P}(\mathcal{K})$. It remains to show that $\mathbb{E}_{y\sim\mathbb{Q}}[\phi_{\text{sig}}(\mathbb{Q}, y)] \le \mathbb{E}_{y\sim\mathbb{Q}}[\phi_{\text{sig}}(\mathbb{P}, y)]$. This means we must have

$$\mathbb{E}_{x\sim\mathbb{P},y\sim\mathbb{Q}}[k_{\text{sig}}(x,y)] \le \frac{1}{2}\mathbb{E}_{x,x'\sim\mathbb{P}}[k_{\text{sig}}(x,x')] + \frac{1}{2}\mathbb{E}_{y,y'\sim\mathbb{Q}}[k_{\text{sig}}(y,y')].$$

Writing $\mathbb{M} = \frac{1}{2}\mathbb{P} + \frac{1}{2}\mathbb{Q}$, a modification of Theorem 2.1 in Berg et al. [BCR84] (pg. 235) gives that

$$\int k_{\text{sig}}(x,y)\, d(\mathbb{P}\otimes\mathbb{Q})(x,y) \le \int k_{\text{sig}}(x,y)\, d(\mathbb{M}\otimes\mathbb{M})(x,y), \tag{11}$$

where $\mathbb{P}\otimes\mathbb{Q}$ denotes the natural product measure on $\mathcal{K}\times\mathcal{K}$. Re-arranging (11), one arrives at the desired result.

To show strict properness, we need to show that $\mathbb{E}_{y\sim\mathbb{Q}}[\phi_{\text{sig}}(\mathbb{Q}, y)] \le \mathbb{E}_{y\sim\mathbb{Q}}[\phi_{\text{sig}}(\mathbb{P}, y)]$ holds with equality iff $\mathbb{P} = \mathbb{Q}$ for all $\mathbb{P}, \mathbb{Q} \in \mathcal{P}(\mathcal{K})$. Suppose that there exists another $\mathbb{P}' \in \mathcal{P}(\mathcal{K})$ such that $\mathbb{E}_{y\sim\mathbb{Q}}[\phi_{\text{sig}}(\mathbb{Q}, y)] = \mathbb{E}_{y\sim\mathbb{Q}}[\phi_{\text{sig}}(\mathbb{P}', y)]$. Then we would have that

$$\mathbb{E}_{x,x'\sim\mathbb{P}'}[k_{\text{sig}}(x,x')] - 2\mathbb{E}_{x\sim\mathbb{P}',y\sim\mathbb{Q}}[k_{\text{sig}}(x,y)] + \mathbb{E}_{y,y'\sim\mathbb{Q}}[k_{\text{sig}}(y,y')] = 0,$$

or that $\mathcal{D}_{k_{\text{sig}}}(\mathbb{P}', \mathbb{Q}) = 0$, which is only true if $\mathbb{P}' = \mathbb{Q}$ due to characteristicness of the kernel $k_{\text{sig}}$.  $\square$

**Proof of Proposition 3.4**

*Proof (Appendix).* The proof follows directly from [GBR+12], Lemma 6. Note that an unbiased estimator for $\mathbb{E}_{x,x'\sim\mathbb{P}}[k_{\text{sig}}(x,x')]$ from i.i.d samples $(x_1,\ldots,x_m), x_i \sim \mathbb{P}$ is given by the U-statistic

$$T_U^1(x_1,\ldots,x_m) = \frac{1}{m(m-1)}\sum_{i\ne j} k_{\text{sig}}(x_i,x_j).$$

Moreover, an unbiased estimate of $\mathbb{E}_{x\sim\mathbb{P}}[k_{\text{sig}}(x,y)]$ is given by

$$T_U^2(x_1,\ldots,x_m,y) = \frac{1}{m}\sum_{i=1}^{m} k_{\text{sig}}(x_i,y).$$

Writing $\hat{\phi}_{\text{sig}}(\mathbb{P}, y) = T_U^1(x_1,\ldots,x_m) - 2T_U^2(x_1,\ldots,x_m,y)$ completes the proof.  $\square$

## B  Experiments

All experiments were run on a NVIDIA GeForce RTX 3070 Ti GPU, except the experiment in Section 4.4 for which the NSPDE model was trained using a NVIDIA A100 40GB GPU.

Here we provide details for each of the experiments outlined in the body of the paper. We also provide some extra methods of evaluation aside from the KS test. These include the following:

1. **Qualitative plot**: We give a plot of samples $\mathbb{P}_{X^\theta}$ from a trained generator against the true data measure $\mathbb{P}_{X^{\text{true}}}$.

2. **Autocorrelation**: To measure temporal dependencies or correlations, we leverage the autocorrelation function

$$\text{ACF}_\ell = \frac{1}{N\sigma^2}\sum_{t=l}^{N}(X_t - \mu)(X_{t-l} - \mu),$$

   where $\mu$ is the average of the path $X_t$ over $[0, N]$ and $\sigma^2$ is the corresponding variance. We provide a qualitative plot of $\text{ACF}_\ell$ for each generator against the real data measure. We also provide a table summarizing the scores for some of the earliest lags $\ell \in \mathbb{N}$.

3. **Cross-correlation**: We provide average cross-correlation scores $(r_t, r_{t,\ell}^2)$ between the returns process associated to $X_t \sim \mathbb{P}_{X^\theta}$ and the squared, lagged returns process $r_{t,\ell}^2$. We present the scores in matrix form. Finally, we provide the MSE between the matrix obtained from $\mathbb{P}_{X^{\text{true}}}$ and those obtained from each generator.

We make a note here that each of the three discriminators performed similarly in the additional quantitative metrics omitted from the body. Finally, we wish to first make the following general notes about each of the three methods studied in this paper:

- **Speed**: Training a Neural SDE with respect to $\phi_{\text{sig}}^N$ was the fastest, followed by the SDE-GAN, and finally with $\phi_{\text{sig}}$ being the slowest. It was possible to decrease training time respect to the latter by using a coarser dyadic refinement in the PDE solver. However, we felt that achieving more accurate gradients at the cost of longer training time was worthwhile.

- **Stability**: The Wasserstein SDE-GAN was the least stable, in terms of the difficulty in obtaining a training instance where the loss converged in reasonable time. Even with fine-tuning of both generator and discriminator parameters, the loss associated to the SDE-GAN tended to oscillate, making obtaining a converged model a very difficult task with the hardware available to us.

- **Scaling**: All of the results in the paper are sensitive to path scalings; moreso with the signature kernel-based approaches, less so with the Wasserstein approach. The basic idea is as follows: the signature kernel-based methods will tend to fail if paths are scaled too low (resulting in lower-order terms dominating the calculation of $k_{\text{sig}}$) or too high (the sum, although finite, can exceed a 64-bit float quite easily). Path scaling (and transformations) form an integral part in training a successful generative model, and we have tried to be as descriptive as possible regarding this matter. The details as to why scalings matter have been touched upon in [CLX21]; we intend to expand upon this in a future work.

- **Standardisation**: On a similar note, standardizing path data before training was often found to improve the stability of training in any setting. By standardization we are referring to transforming each marginal of paths $X \sim \mathbb{P}_{X^{\text{true}}}$ via the transformation $\hat{X}_t = (X_t - \mu_T)/\sigma_T$, where $\mu_T = \mathbb{E}_{\mathbb{P}_{X^{\text{true}}}}[X_T]$ and $\sigma_T = \mathbb{E}_{\mathbb{P}_{X^{\text{true}}}}[(X_T - \mu_T)^2]$. By having the terminal marginal distributed standard normal, the task of finding suitable path scalings and smoothing parameters in the RBF kernel was made much simpler, as this task became less problem-specific.

## B.1 Geometric Brownian motion

**Data processing and hyperparameters** To generate our data measure, we simulate $32768$ paths according to eq. (9) using the `torchsde` package. These were solved over the interval $[0, 64]$ by setting $y_0 = 1, \mu = 0, \sigma = 0.2$, with $dt = 0.1$. Paths were then interpolated along the grid $\Delta = \{0, 1, 2, \ldots, 63\}$, so each element of the training set had total length $64$. Stochastic integrals were taken in the Itô sense and the driving noise $W$ was taken in the general sense. We used the SRK method to solve the corresponding SDE. Each path is time-augmented, so $\hat{X}_t = (t, X_t)$ at each point on the grid. After we have simulated our dataset, we standardized each path as outlined in the dot points above.

**Generator hyperparameters** The generator is a Neural SDE with vector fields $\mu_\theta : [0, T] \times \mathbb{R}^y \to \mathbb{R}^y$ and $\sigma_\theta : [0, T] \times \mathbb{R}^{y \times w} \to \mathbb{R}^y$ taken to be neural networks with 1 hidden layer, and 16 neurons in said layer. As per [KFL+21] the LipSwish activation function was used to ensure the Lipschitz condition held on the vector fields of the Neural SDE. We also used the final tanh regularisation which we found was necessary for training success. Thus we have that

$$\mu_\theta, \sigma_\theta \in \mathcal{NN}(1, 16, 1, \text{LipSwish}, \tanh).$$

The size of the hidden state of the neural SDE was chosen to be $y = 8$, and the noise dimension was chosen to be $w = 3$. Stochastic integration was taken in the Itô sense and we set $dt = 1$ over $[0, 63]$. As we are not learning an initial distribution in this instance, we modified the generator architecture to have $\xi_\theta(X_0) = a$ for some $a \in \mathbb{R}$, where $\xi_\theta$ is the network acting on the initial condition. Before passing to the discriminator, both generated and real paths were translated to start at $0$.

**Discriminator hyperparameters** For the signature kernel-based discriminators, we applied the time normalisation transformation so the time component of both the real and generated paths was over $[0, 1]$ as opposed to $[0, 63]$. This was to ensure each channel of the generated and real data evolved over a similar scale. For training with respect to $\phi_{\text{sig}}$, we set the order of dyadic refinement associated to the PDE solver for the signature kernel to 1. We also used three different kernels, corresponding to three different scalings of the paths, for increased expressivity. For $\phi_{\text{sig}}^N$, we set the order of truncation equal to $N = 3$. Finally, for the SDE-GAN, we chose the drift and diffusion vector fields to be feed-forward neural networks

$$f_\phi, g_\phi \in \mathcal{NN}(1, 16, 1, \text{LipSwish}, \tanh),$$

matching that from the generator.

**Training hyperparameters** All methods used a batch size of 128 and the Adam optimisation algorithm for backpropagating through the generator optimisers, except for the SDE-GAN, which as suggested by the authors we used Adadelta. As a remark, we did not see much difference in using either Adam, Adadelta, or RMSProp, although we did see poorer performance using pure SGD, with or without momentum. Learning rates were roughly proportional to the average size of the batched loss: as a rough guide, proportionality like $\eta_G \times \mathcal{L}(\theta) \approx 10^{-5}$ tended to yield good results, with the generator learning rate being around $\eta_G \approx 10^{-4}$ for the signature kernel(s), and $\eta_D \approx \eta_G \times 10^{-2}$ for the SDE-GAN. As mentioned in the body, we trained for 4000 steps with $\phi_{\text{sig}}$, 10000 with $\phi_{\text{sig}}^N$ and 5000 with the SDE-GAN to normalise for training time.

**Results** We begin with a qualitative plot of the results from each generator.

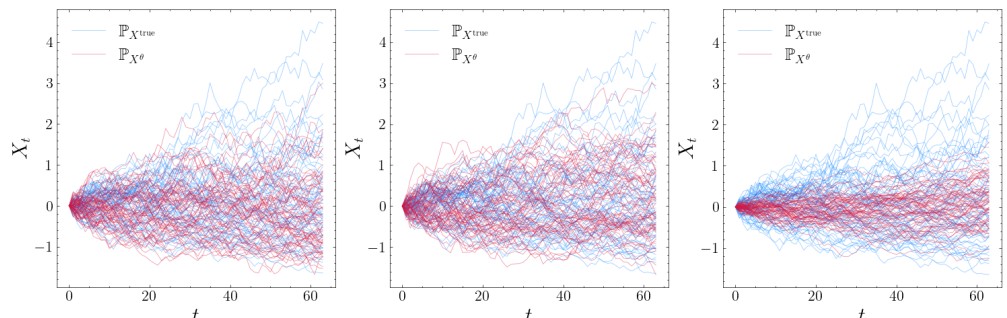

Figure 4: Qualitative plot of generator output versus data measure. Trained with (from left to right): $\phi_{\text{sig}}$, $\phi_{\text{sig}}^N$, SDE-GAN

Table 5 gives the autocorrelation scores for the first five lags for each of the three models, along with plots of the mean ACF values in Figure 5 and the associated $95\%$ confidence intervals. We can see that all three models do well at capturing temporal effects, with the Neural SDE trained with respect to $\phi_{\text{sig}}$ most closely matching the data measure, except in the first lag.

| Discriminator | Lags | | | | |
| --- | --- | --- | --- | --- | --- |
| | $l = 1$ | $l = 2$ | $l = 3$ | $l = 4$ | $l = 5$ |
| SDE-GAN | $\mathbf{0.887 \pm 0.120}$ | $0.782 \pm 0.211$ | $0.686 \pm 0.282$ | $0.597 \pm 0.342$ | $0.515 \pm 0.384$ |
| $\phi_{\text{sig}}^N$ ($N = 3$) | $0.883 \pm 0.115$ | $0.781 \pm 0.197$ | $0.684 \pm 0.261$ | $0.594 \pm 0.320$ | $0.514 \pm 0.364$ |
| $\phi_{\text{sig}}$ | $0.886 \pm 0.111$ | $\mathbf{0.785 \pm 0.199}$ | $\mathbf{0.696 \pm 0.267}$ | $\mathbf{0.612 \pm 0.315}$ | $\mathbf{0.535 \pm 0.350}$ |
| *Data measure* | $0.892 \pm 0.105$ | $0.793 \pm 0.183$ | $0.702 \pm 0.258$ | $0.616 \pm 0.319$ | $0.532 \pm 0.374$ |

Table 5: Sample autocorrelation scores, gBm

Finally, we present the cross-correlation matrices between the returns process $r_t$ and the lagged squared returns process $r_{t-l}^2$ for lags $l = \{0, 1, 2, 3, 4, 5\}$. Again all models tend to perform quite well in that they match relational dynamics observed in the data measure. Table 6 gives the MSE between the generated matrices and the data matrix. We see that the Neural SDE trained with $\phi_{\text{sig}}$

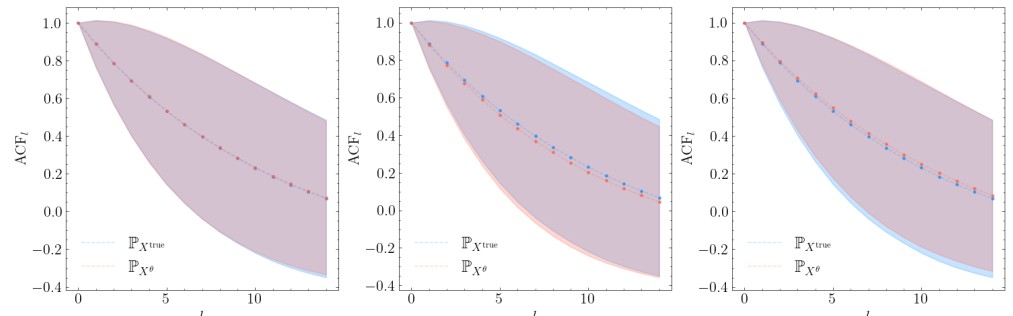

Figure 5: Qualitative plot of ACF scores, generator output versus data measure. Trained with (from left to right): $\phi_{\text{sig}}$, $\phi_{\text{sig}}^N$, SDE-GAN

achieves the lowest score of the three, however again performance is strong regardless of method used for training.

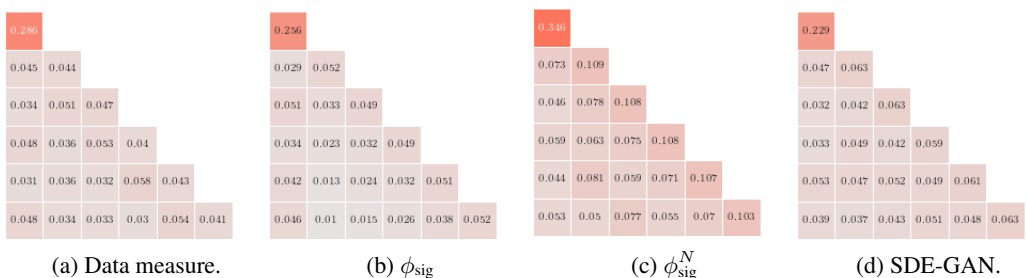

Figure 6: Cross-correlation matrices, gBm

| Discriminator | MSE |
|---|---|
| SDE-GAN | 0.014688 |
| $\phi_{\text{sig}}^N$ ($N=3$) | 0.066745 |
| $\phi_{\text{sig}}$ | **0.010718** |

Table 6: MSE between cross-correlation matrices, gBm

## B.2 Rough Bergomi

**Data processing and hyperparameters** We simulate $32768$ paths to make up our data measure via the `rBergomi` Python package[5]. We fixed the time window to be $[0, 2]$, and specified $dt = 1/32$, so paths were of length $64$. We chose $(\xi_0, \eta, \rho, H) = (0.04, 1.5, -0.7, 0.2)$ and set $d = 1$. Paths started at $1$. As always, paths were time-augmented. Paths were normalised to start at $0$ via translation and were standardized again according to the terminal data from the train set. A final point is that although the data was generated over $[0, 2]$, the time grid passed to the generators in an optimiser step was $\Delta = \{0, 1, \dots, 63\}$. We found that this improved performance.

**Generator hyperparameters** Given the increased complexity of the data generating model, we increased the expressivity of the vector fields governing the drift and diffusion vector fields $\mu_\theta$ and $\sigma_\theta$. This was done by increasing the depth and width of the constituent feed-forward networks to include 3 hidden layers of size 32. We also increased the size of the hidden state to $y = 16$ and the

---

[5]See `https://github.com/ryanmccrickerd/rough_bergomi`

noise dimension to $w = 8$. Thus

$$\mu_\theta \in \mathcal{NN}(17, 32, 32, 32, 16; \text{LipSwish}, \text{LipSwish}, \text{LipSwish}, \tanh)$$

and

$$\sigma_\theta \in \mathcal{NN}(17, 32, 32, 32, 128; \text{LipSwish}, \text{LipSwish}, \text{LipSwish}, \tanh).$$

**Discriminator hyperparameters**    For training with respect to $\phi_{\text{sig}}$, we mapped path state values to $(\mathcal{H}, \kappa)$ where $\kappa$ denotes the RBF kernel on $\mathbb{R}^2$. We set associated the smoothing parameter $\sigma = 1$. For $\phi_{\text{sig}}^N$, we increased the truncation level to $N = 5$. In both these settings we again applied the time normalisation transformation on both the generated and data measure paths before being passed through the loss function. For the SDE-GAN, we increased the expressiveness of the vector fields governing the Neural CDE in the same way as we did the Neural SDE.

**Training hyperparameters**    Learning rates for the Neural SDE trained according to $\phi_{\text{sig}}$ was set to $\eta_G = 1 \times 10^{-4}$. Due to the increasing number of terms in the expected signature for the truncated MMD approach, we had to reduce the learning rate to $\eta_G = 1 \times 10^{-6}$ - larger values caused instability in the training procedure. The SDE-GAN was again quite difficult to train, however we were able to have some success by setting $\eta_D \approx 2 \times 10^{-3}$ and $\eta_G \approx 1 \times 10^{-3}$. Initialisation of the generator vector fields was especially important for the Wasserstein method, as initialisation too far from the data measure cause oscillatory patterns in the training loss, which leads to more epochs required for the loss to converge. We again used the Adam optimisation algorithm for the MMD-based discriminator/generators and Adadelta for the SDE-GAN. We trained for the same number of steps as per the gBm method.

**Results**    Figure 7 gives a qualitative plot of the simulated paths.

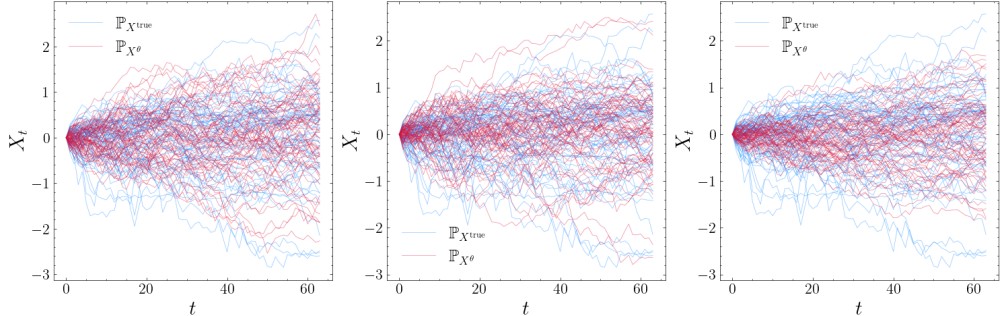

Figure 7: Qualitative plot of generator output versus data measure. Trained with (from left to right): $\phi_{\text{sig}}, \phi_{\text{sig}}^N$, SDE-GAN

Figure 8 gives the same plot of the ACF scores at corresponding lags for the data measure and each of the generated models, along with the 95% confidence interval. Table 7 explicitly gives these scores.

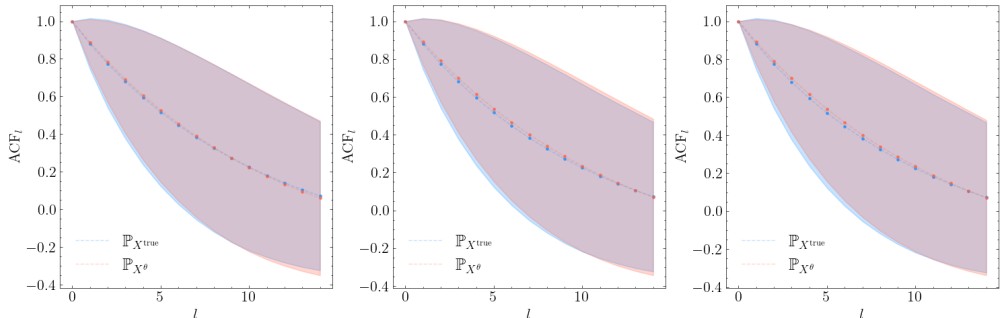

Figure 8: Qualitative plot of ACF scores, generator output versus data measure. Trained with (from left to right): $\phi_{\text{sig}}, \phi_{\text{sig}}^N$, SDE-GAN

| Discriminator | | | Lags | | |
| --- | --- | --- | --- | --- | --- |
| | $l = 1$ | $l = 2$ | $l = 3$ | $l = 4$ | $l = 5$ |
| SDE-GAN | $0.893 \pm 0.105$ | $0.795 \pm 0.191$ | $0.705 \pm 0.262$ | $0.622 \pm 0.319$ | $0.543 \pm 0.377$ |
| $\phi_{\mathrm{sig}}^{N}\,(N = 5)$ | $0.897 \pm 0.115$ | $0.799 \pm 0.208$ | $0.710 \pm 0.289$ | $0.628 \pm 0.338$ | $0.546 \pm 0.383$ |
| $\phi_{\mathrm{sig}}$ | $\mathbf{0.890 \pm 0.115}$ | $\mathbf{0.790 \pm 0.203}$ | $\mathbf{0.702 \pm 0.269}$ | $\mathbf{0.618 \pm 0.316}$ | $\mathbf{0.521 \pm 0.382}$ |
| *Data measure* | $0.885 \pm 0.121$ | $0.778 \pm 0.215$ | $0.685 \pm 0.278$ | $0.600 \pm 0.336$ | $0.521 \pm 0.380$ |

Table 7: Sample autocorrelation scores, rBergomi

Finally, we present the same cross-correlation matrices, along with the MSE between either of the three generators and the data measure. Although each of the models perform well, the generator trained with $\phi_{\mathrm{sig}}$ achieves the best results.

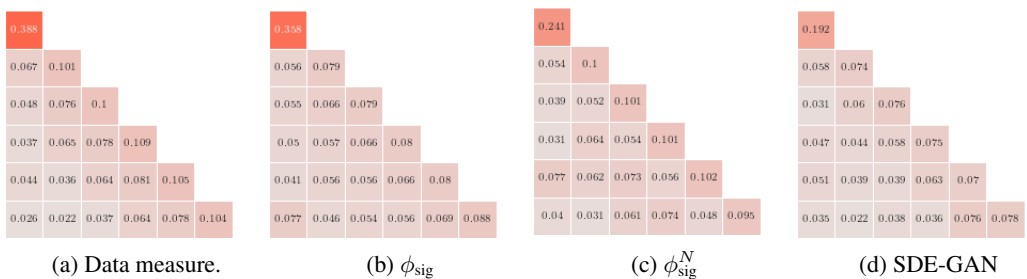

(a) Data measure.  (b) $\phi_{\mathrm{sig}}$  (c) $\phi_{\mathrm{sig}}^{N}$  (d) SDE-GAN

Figure 9: Cross-correlation matrices, rBergomi

| Discriminator | MSE |
| --- | --- |
| SDE-GAN | 0.091707 |
| $\phi_{\mathrm{sig}}^{N}\,(N = 5)$ | 0.054731 |
| $\phi_{\mathrm{sig}}$ | **0.016785** |

Table 8: MSE between cross-correlation matrices, rBergomi

### B.3 Multidimensional real data

We now give the details regarding the unconditional generation of foreign exchange data.

**Data processing and hyperparameters**  Data is given by hourly returns associated to the currency pairs EUR/USD and USD/JPY. We stride the concatenated time series (corresponding to close prices at each time) into paths of length $\ell = 64$. Paths are normalized to start at $1$. We augmented the state values with their original timestamps, (as epoch seconds). Call this dataset $\mathcal{Y}$. As we are dealing with financial data, one can expect the time intervals between prices to be irregular. This is not usually an issue when using signature methods. However, in this setting we found training to be less stable if paths were not normalised to evolve over the same time grid as that which the neural SDE was solved over in a forward pass.

To circumvent this issue, for every $y \in \mathcal{Y}$ we find the median terminal time $\tilde{T}$, where

$$\tilde{T} = \mathrm{Median}_{i=1,\ldots,|\mathcal{Y}|}[t_\ell^i / t_0^i].$$

All paths whose terminal time is greater than $\tilde{T}$ were filtered out of the dataset which we call $\tilde{\mathcal{Y}}$. We then define an evenly-spaced time grid $\Delta^* = \{0, \ldots, \tilde{T}\}$ containing 64 observations in total, and linearly interpolate each $y \in \tilde{\mathcal{Y}}$ over this grid, where we use these interpolated coefficients to

form our train and test sets. Again we standardize using the terminal values of the train set data. We simulate our generators over the time grid $\Delta = \{0, \dots, 63\}$ as we found that using $\Delta^*$, the realistic time-grid (in fractions of a year) induced little variability in the generated paths; i.e., the quadratic variation associated to the generated paths was significantly lower than that obtained from the real data measure.

**Generator hyperparameters** The generator maintains the same architecture as outlined in the rBergomi section. We tried increasing the size of the hidden state to $x = 32$ and the noise state $w = 16$ but found that this had little impact on training performance. We also found that increasing the expressivity of the neural vector fields did not overly impact performance; neither refining the mesh over which the Neural SDE was solved.

**Discriminator hyperparameters** We used the same discriminator hyperparameters for each of the three methods as per the rBergomi section.

**Training hyperparameters** We used the same batch size (128) as per the previous sections. We allowed for increased training time here, training with respect $\phi_{\text{sig}}$ for 4000 steps, 15000 for $\phi_{\text{sig}}^N$ and 10000 for the SDE-GAN. The same learning rate parameters were used as well. We did not use any learning rate annealers. The Adam optimisation algorithm was employed for the MMD-based generators, whereas again Adadelta was used for the Wasserstein case.

**Results** Extended results are provided as per the previous sections. We being with the qualitative plots.

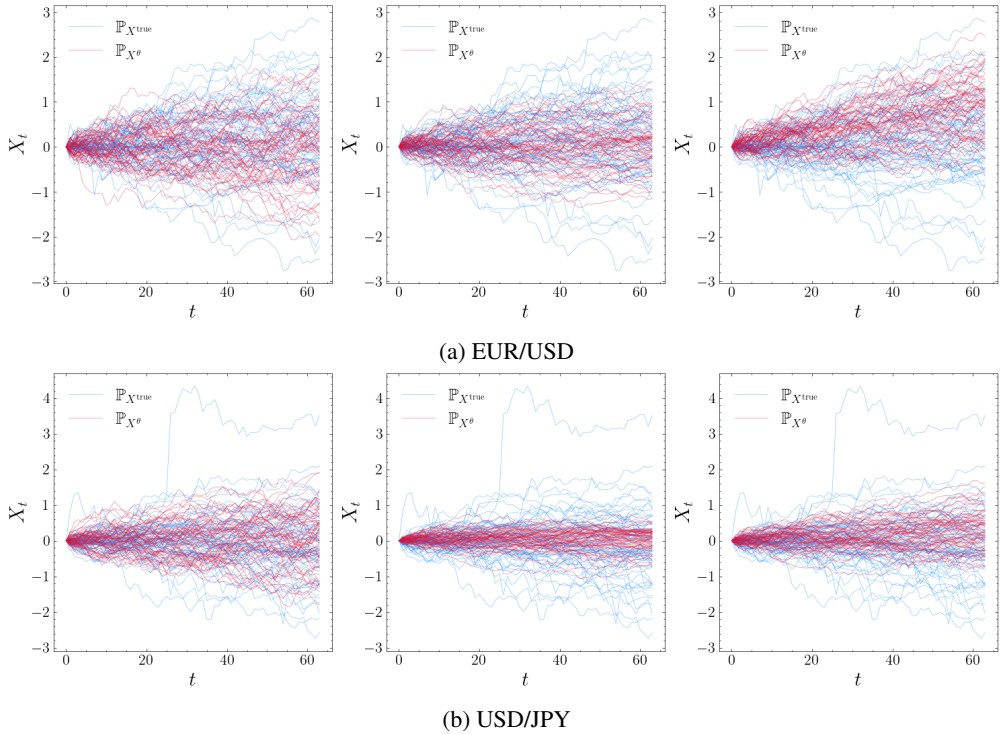

(a) EUR/USD

(b) USD/JPY

Figure 10: Qualitative plot of generator output versus data measure. Trained with (from left to right): $\phi_{\text{sig}}$, $\phi_{\text{sig}}^N$, SDE-GAN

Visual inspection gives that the generator trained with respect to $\phi_{\text{sig}}$ appears to have most accurately captured the data measure, in particular the less regular, outlier paths. In contrast the SDE-GAN and truncated kernel methods tend to over-represent the mean element. For the GAN this could be the "mode collapse" phenomenon in effect, whereas in the case of the Neural SDE trained with $\phi_{\text{sig}}^N$, it is likely that higher-order terms cannot be discarded if one wishes to accurately model the data measure.

We now provide the plot associated to the ACF scores obtained from training with respect to each generator, along with the summarizing table. Table 9 shows that that each of the discriminators perform relatively well, aside from the EURUSD autocorrelative factors obtained via traning the Neural SDE in the SDE-GAN framework. Finally we give the cross-correlation matrices and the associated MSEs. The Neural SDE trained with respect to $\phi_{\text{sig}}$ appears to perform the best by this evaluation metric.

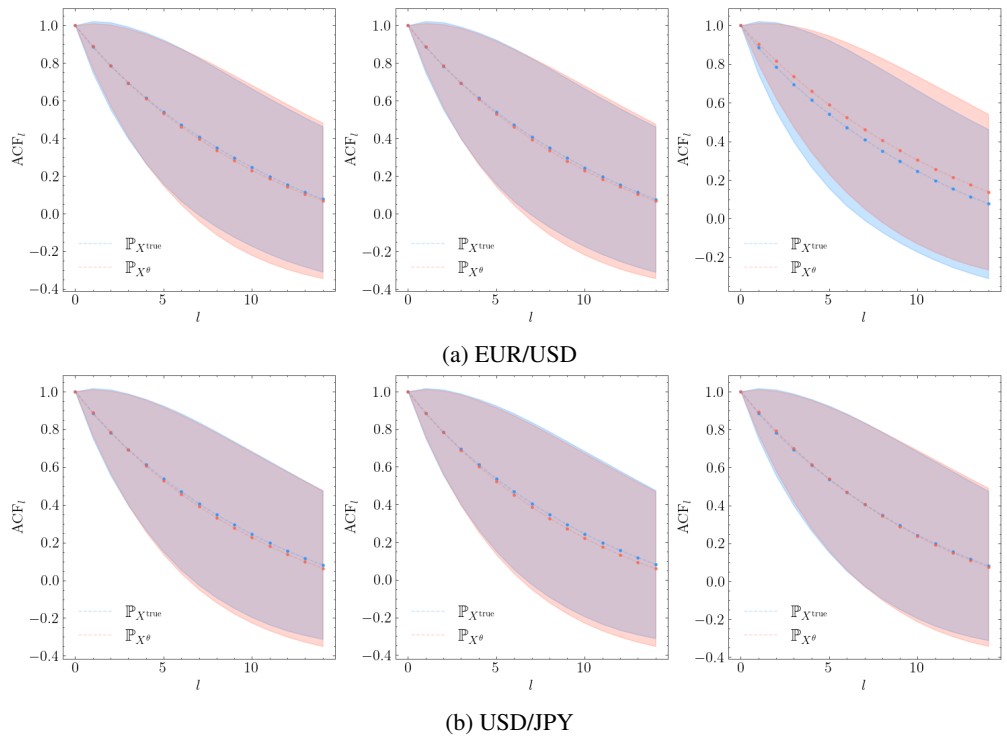

(a) EUR/USD

(b) USD/JPY

Figure 11: Qualitative plot of ACF scores, generator output versus data measure. Trained with (from left to right): $\phi_{\text{sig}}$, $\phi_{\text{sig}}^N$, SDE-GAN

| Discriminator | Lags | | | | |
| | $l=1$ | $l=2$ | $l=3$ | $l=4$ | $l=5$ |
|---|---|---|---|---|---|
| SDE-GAN | $0.905 \pm 0.108$ | $0.817 \pm 0.195$ | $0.736 \pm 0.264$ | $0.660 \pm 0.319$ | $0.590 \pm 0.362$ |
| $\phi_{\text{sig}}^N$ ($N=5$) | $\mathbf{0.889 \pm 0.123}$ | $\mathbf{0.788 \pm 0.215}$ | $\mathbf{0.695 \pm 0.289}$ | $0.610 \pm 0.345$ | $0.532 \pm 0.388$ |
| $\phi_{\text{sig}}$ | $0.890 \pm 0.123$ | $0.790 \pm 0.218$ | $0.699 \pm 0.291$ | $\mathbf{0.615 \pm 0.347}$ | $\mathbf{0.538 \pm 0.388}$ |
| *Data measure* | $0.885 \pm 0.140$ | $0.785 \pm 0.236$ | $0.696 \pm 0.302$ | $0.615 \pm 0.302$ | $0.541 \pm 0.387$ |

(a) EUR/USD

| Discriminator | Lags | | | | |
| | $l=1$ | $l=2$ | $l=3$ | $l=4$ | $l=5$ |
|---|---|---|---|---|---|
| SDE-GAN | $0.891 \pm 0.121$ | $0.791 \pm 0.216$ | $0.700 \pm 0.290$ | $0.616 \pm 0.346$ | $\mathbf{0.539 \pm 0.389}$ |
| $\phi_{\text{sig}}^N$ ($N=5$) | $\mathbf{0.887 \pm 0.123}$ | $\mathbf{0.785 \pm 0.218}$ | $\mathbf{0.692 \pm 0.292}$ | $0.607 \pm 0.348$ | $0.529 \pm 0.390$ |
| $\phi_{\text{sig}}$ | $0.891 \pm 0.121$ | $0.790 \pm 0.215$ | $0.698 \pm 0.288$ | $\mathbf{0.614 \pm 0.344}$ | $0.537 \pm 0.385$ |
| *Data measure* | $0.885 \pm 0.132$ | $0.784 \pm 0.227$ | $0.694 \pm 0.296$ | $0.613 \pm 0.347$ | $0.539 \pm 0.385$ |

(b) USD/JPY.

Table 9: Sample autocorrelation scores, forign exchange data

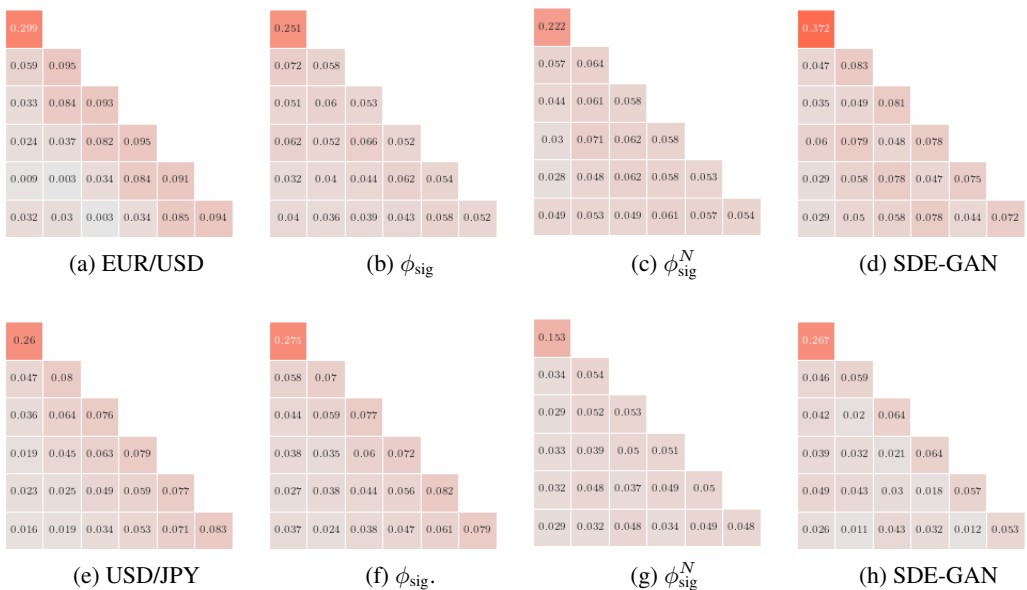

Figure 12: Cross-correlation matrices, foreign exchange data

| Discriminator | MSE | |
| --- | --- | --- |
| | EUR/USD | USD/JPY |
| SDE-GAN | 0.051728 | 0.027539 |
| $\phi_{\text{sig}}^N$ ($N = 5$) | 0.045966 | 0.036628 |
| $\phi_{\text{sig}}$ | **0.035791** | **0.003898** |

Table 10: MSE between cross-correlation matrices, foreign exchange data

## B.4 Conditional generation

In this section we describe in detail the training procedure for the conditional generator.

**Problem setting** The conditioning variables are given by path segments $x : [t_0 - dt, t_0] \to \mathbb{R}^2$, representing time-augmented asset price values. At time $t_0$, one wishes to make a prediction about the resultant path $y : [t_0, t_0 + dt'] \to \mathbb{R}^2$ conditional on $x$. Here, $dt, dt'$ are hyperparameters describing how much of the past one wishes to consider and how far into the future one wishes to forecast. With $x \sim \mathbb{Q}$ We thus want to train a conditional generator so that $\mathbb{P}_{X^\theta}(\cdot|x) = \mathbb{P}_{X^{\text{real}}}(\cdot|x)$. We briefly state the three major difficulties associated to this generation problem:

1. **Unobservable true conditional distribution**. In practice, one never observes the entire true conditional distribution $\mathbb{P}_{X^{\text{real}}}(\cdot|x)$: only a sample from it. This means that classical metrics on path space (MMD, Wasserstein, and so on) cannot be used without modification, or making assumptions about the relationship between the conditioning and resultant paths.

2. **Using paths as conditioning variables**. It is not immediately clear what is the best way to consume a path as a conditioning variable for a given generator.

3. **"Unseen" conditioning variables**. It is not guaranteed that the conditional generator will behave in an expected way if an as-yet unseen conditioning variable is provided (by unseen, we are referring to within the training procedure). These conditioning variables are often the ones of interest.

Our procedure attempts to solve the first and second problems with our architecture choices on the conditional generator, and our choice of loss function. The third issue is omnipresent in conditional modelling.

The generator now is given by a (conditional) Neural SDE with architecture given by

$$Y_0 = \xi_\theta(x_{t_0}, C(x)), \quad dY_t = \mu_\theta(t, Y_t, C(x))dt + \sigma_\theta(t, Y_t, C(x)) \circ dW_t, \quad X_t = \pi_\theta(Y_t, C(x))$$

for $\mu_\theta : [t_0, t_0+dt'] \times \mathbb{R}^y \times \mathbb{R}^{d_C} \to \mathbb{R}^y, \sigma_\theta : [t_0, t_0+dt'] \times \mathbb{R}^x \times \mathbb{R}^{d_C} \to \mathbb{R}^{x \times w}, \xi_\theta : \mathbb{R}^x \times \mathbb{R}^{d_C} \to \mathbb{R}^y$, and $\pi_\theta : \mathbb{R}^y \times \mathbb{R}^{d_C} \to \mathbb{R}^x$. Here, $C(x)$ denotes the function acting on the conditioning path and encoding it as a vector in $\mathbb{R}^{d_C}$. A natural way to perform this encoding is via the truncated signature $S^M(X)$ of the path $x$. In this way the neural networks defining the vector fields in the generator (for instance) are now mappings

$$\mu_\theta : [t_0, t_0 + dt'] \times \mathbb{R}^y \times \mathbb{R}^{1+d+d^2+\cdots+d^N} \to \mathbb{R}^y,$$
$$\sigma_\theta : [t_0, t_0 + dt'] \times \mathbb{R}^y \times \mathbb{R}^{1+d+d^2+\cdots+d^N} \to \mathbb{R}^{y \times w}.$$

We note here that all of the regularity conditions required to ensure a strong solution to the standard Neural SDE here remain satisfied; we are only augmenting each of the trainable components to accept the encoded conditioning path. We also note here that this technique is flexible enough to include any amount of $\mathbb{R}^d$−valued conditioning variables.

**Data processing and hyperparameters**   Data comes from 15-minute close prices associated to the EUR/USD price pair. We again extracted paths of length 48 (normalising for erroneous terminal times as per the unconditional setting) and split these paths into conditioning-resultant pairs $\{x^i, y^i\}_{i=1}^N$ with $x^i$ representing the first 32 observations and $y^i$ the next 16. We normalized both sets of paths by their initial value. Instead of standardizing, in this setting we scaled all path values up by a factor of 100. We found this was crucial so that the lower-order signature terms did not overly contribute to the value of the signature kernel. Both conditioning and resultant paths were then translated to start at 0. In total the dataset size was comprised of 52428 conditioning/resultant pairs.

**Generator hyperparameters**   The generator is a conditional Neural SDE. Stochastic integration was again taken in the Itô sense and we used the Euler method. The noise size was set to $w = 8$, the size of the hidden state was taken $y = 16$. The MLPs governing the vector fields were the same as per the rBergomi and multidimensional unconditional examples, except we increased the width of the layers in the neural networks to 64 neurons. We conditionalized the input paths via the truncated log-signature of order 5. In order to estimate the batched loss, we need to specify the size of the conditional distribution $\mathbb{P}_{X^\theta}(\cdot|x)$ output by the conditional generator, which we set to 32 paths. Finally, we applied the time normalisation and lead-lag transformations to the input paths $x$ before taking their truncated log-signature.

**Discriminator hyperparameters**   We trained with respect to $\phi_{\text{sig}}$. Again we lifted paths via the RBF kernel and chose the smoothing parameter $\sigma = 1$. We used a dyadic refinement level of 1 for the PDE solver associated to $k_{\text{sig}}$. All paths had the time normalisation transformation applied to them before having the loss evaluated.

**Training hyperparameters**   We set the batch size equal to 128 and trained the conditional generator for 10000 steps. We set the learning rate $\eta_G = 2 \times 10^{-6}$ and used the Adam optimisation algorithm in PyTorch. No learning rate annelears were used.

**Results**   Results are presented in the body of the paper.

### B.5   Simulation of limit order books

The Neural SPDE model introduced in [SLG22] extends Neural SDEs to model spatiotemporal dynamics by parametrising the differential operator, drift and diffusion of SPDEs of the type

$$dY_t = (\mathcal{L}Y_t + \mu(Y_t))dt + \sigma(Y_t)dW_t \tag{12}$$

where both $\mu$ and $\sigma$ are local operators acting on the function $Y_t$ that is, $\mu(Y_t)(x)$ and $\sigma(Y_t)(x)$ only depend on $Y_t(x)$. Moreover, it is assumed that $\mathcal{L}$ is a linear differential operator generating a semigroup $e^{t\mathcal{L}}$ which can be written as a convolution with a kernel $\mathcal{K}_t$.

Let $D \subset \mathbb{R}^d$ be a bounded domain. Let $W : [0, T] \to L^2(D, \mathbb{R}^{d_w})$ be a Wiener process and $a$ an $L^2(D, \mathbb{R}^{d_a})$-valued Gaussian random variable. The values $d_w, d_a \in \mathbb{N}$ are hyperparameters describing the size of the noise. A Neural SPDE is a model of the form

$$Y_0(x) = \ell_\theta(a(x)), \quad Y_t = \mathcal{K}_t * Y_0 + \int_0^t \mathcal{K}_{t-s} * (\mu_\theta(Y_s) + \sigma_\theta(Y_s)\dot{W}_s^\epsilon)ds, \quad X_t^\theta(x) = \pi_\theta(Y_t(x)).$$

for $t \in [0, T]$ and $x \in D$ where $Y : [0, T] \to L^2(D, \mathbb{R}^{d_y})$ is the mild solution, if it exists to the SPDE in Equation (12) with regularised driving noise $W^\epsilon$ and where $*$ denotes the convolution in space with the kernel $\mathcal{K}_t : D \times D \to \mathbb{R}^{d_y \times d_y}$ (see [SLG22] for more details). Similarly to the Neural SDE model,

$$\ell_\theta : \mathbb{R}^{d_a} \to \mathbb{R}^{d_y}, \quad \mu_\theta : \mathbb{R}^{d_y} \to \mathbb{R}^{d_y}, \quad \sigma_\theta : \mathbb{R}^{d_y} \to \mathbb{R}^{d_y \times d_w}, \quad \pi_\theta : \mathbb{R}^{d_y} \to \mathbb{R}^{d_x}$$

are feedforward neural networks. Imposing globally Lipschitz conditions (by using ReLU or tanh activation functions in the neural networks $\mu_\theta$ and $\sigma_\theta$) ensures the existence and uniqueness of the mild solution $Y$. Finally, we note that in [SLG22], the authors propose two distinct algorithms to evaluate the Neural SPDE model based on two different parameterisations of the kernel $\mathcal{K}$.

Next, we provide more details on how we trained such a Neural SPDE model to generate Limit Order Book (LOB) dynamics [GPW+13]. The increasing availability of LOB data has instigated a significant interest in the development of statistical models for LOB dynamics. In recent years, new models based on SPDEs have been proposed to accurately describe and analyse these complex dynamics [HKN20, CM21].

**Data processing and hyperparameters**   We used real LOB data from the NASDAQ public exchange made publicly available in [NMK+18] which consists of about 4 million timestamped events over 10 consecutive trading days with $L = 10$ price levels on each side (bid and ask) of the LOB. Three versions of this dataset are provided, each normalised using a different technique. We used the data normalised with z-scores and split the LOB trace into sub-traces of length $T = 30$.

**Generator hyperparameters**   The generator is a Neural SPDE driven by a cylindrical Wiener process $W$ with $d_w = 2$. The vector fields $\mu_\theta$ and $\sigma_\theta$ are taken to be single layer perceptrons with $d_y \in \{16, 32\}$ followed by batch normalization and tanh activation function. Thus we obtain

$$\mu_\theta \in \mathcal{NN}(d_h, d_h, \text{BatchNorm}, \text{tanh}), \quad \sigma_\theta \in \mathcal{NN}(d_h, d_h \times d_w, \text{BatchNorm}, \text{tanh})$$

We used the second evaluation method proposed in [SLG22, Section 3.3] with 4 Picard's iterations and maximum number of frequency modes in $\{10, 20\}$ in the spatial direction and fixed to 20 in the temporal direction. Instead of sampling the initial condition $a$ from a $L^2(D, \mathbb{R}^{d_a})$-valued Gaussian, we simply used the samples from $X_0^{\text{true}}$, in which case $d_a = 1$.

**Discriminator hyperparameters**   We integrated in time the output trajectories from the generator, as we observed this yielded more stable kernel scores. We mapped the path state values into $\mathcal{H}_\kappa$ where $\kappa$ denotes a SE-T kernel on $L^2(D)$ with $D = [0, 1]$, that is, a kernel defined for all $f, g \in L^2(D)$ by $\kappa(f, g) = e^{-\frac{1}{2\sigma^2}\|T(f)-T(g)\|_{\mathcal{Y}}^2}$ where $T : L^2(D) \to \mathcal{Y}$ is a Borel measurable, continuous and injective map. We considered three SE-T kernels respectively termed ID, SQR and CEXP:

1. (ID) SE-T kernel with $T : L^2(D) \to L^2(D)$ defined for all $f \in L^2(D)$ by $T(f) = f$
2. (SQR) SE-T kernel with $T : L^2(D) \to L^2(D) \oplus L^2(D)$ defined by $T(f) = (f, f^2)$
3. (CEXP) SE-T kernel with $T : L^2(D) \to L^2(D)$ defined by $T(f) = C_{F,l}(f)$ where $C_{F,l}$ is the covariance operator associated to the kernel $k_{F,l}$ defined for all $x, x' \in D$ by

$$k_{F,l}(x, x') = e^{-\frac{1}{2l^2}(x-x')^2} \sum_{n=0}^{F-1} \cos(2\pi n(x - x'))$$

For ID and SQR, we used $\sigma \in \{1, 10\}$, and for CEXP we used $(\sigma, l, F) \in \{(1, 1, 5), (10, 10, 5)\}$. We then used a dyadic order of 1 to compute the signature kernel.

**Training hyperparameters**    We set the learning rate of the generator $\eta_G$ to be $1 \times 10^{-3}$ and trained it for a maximum number of $1\,500$ epochs. We used a batch size of $64$ due to memory constraints and the Adam optimizer with the default parameters of PyTorch.

