# OpenReview forum: "Non-adversarial training of Neural SDEs with signature kernel scores"
_NeurIPS.cc/2023/Conference — NeurIPS 2023 poster_

### Official Review · Reviewer_dxEe · 2023-07-04

**Soundness:** 3 good
**Presentation:** 3 good
**Contribution:** 3 good
**Rating:** 5
**Confidence:** 1

**Summary:**

This paper proposes a signature kernel-based Neural SDE by using signature kernel scores. The proposed method eliminates existing state-of-the-art methods' mode collapse and instability using adversarial training.

**Strengths:**

1. In all experiments, the proposed method outperforms existing methods.
2. The explanation in chapter three is clear and reproducible because the code is also provided.


**Weaknesses:**

1. This paper's main contribution includes using signature kernel scores for the discriminator, but there needs to be more motivated to use signature kernels.
2. There is no sub-caption for each of Figure 1, so it is unclear which generator it is.


**Questions:**

1. Is that why you chose the signature kernel among the various kernels? Also, if different kernels are adaptable, I would like to know the results of your experiments with other kernels.
2. Which generator does the sub Figure in Figure 1 correspond to?

Minor corrections

3. The unconditional_nsde.ipynb included in the code provided need to be corrected?
```
#cond_                    = data_type == "rBergomi"
cond_                    = data_type == "gbm"
```

**Limitations:**

yes

---

> ### Author Rebuttal · Authors · 2023-08-09
>
> **Contribution**
>
> Please see the response to Reviewer xQxq under the header **Significance**, where we explain that signature kernels allow us to introduce a class of scoring rules for infinite-dimensional spaces of paths, adaptable to spatiotemporal signals, and with strict properness and consistency guarantees.
>
> **Figure 1**
>
> In the preceding paragraph, we explain that the three subfigures are obtained by training Neural SDEs with three different discriminators. We agree with you that the caption should be expanded to clarify which histogram belongs to which discriminator, and we will amend the caption accordingly.
>
> **Minor corrections**
>
> Although this line does not affect the output of the code as it is a commented line, we appreciate that it can be confusing for the reader. The line was left to help guide the user on how to choose a different generation problem. We are happy to remove it.

---

> > ### Comment · Reviewer_dxEe · 2023-08-19
> >
> > Thank you for the rebuttal. All my questions have been answered. I also read other reviews and authors' responses and decided to keep my initial rating.

---

### Official Review · Reviewer_kdBJ · 2023-07-07

**Soundness:** 3 good
**Presentation:** 3 good
**Contribution:** 3 good
**Rating:** 6
**Confidence:** 4

**Summary:**

In this work, the authors proposed to generate time-series using neural stochastic differential equations using the scoring rules-based training objective which is on signature paths computed from signature kernels. Combining the idea of generative adversarial network, this work also employ the generator-discriminator pair, but solve them as a system of linear PDEs where the adjoint method is applied to reduce the memory cost. Experiments are conducted accordingly on sequential data, including financial datasets.

**Strengths:**

1. This paper is written in high clarity. All formulations and derivations are clearly elaborated.

2. The idea is promising and novel to combine generative adversarial networks and neural stochastic differential equations, avoiding adversarial training which can cause instability in training.

3. Rough path theory is a relatively new topic in mathematics. It has penetrated in the machine learning community since only 2019. It is interesting to utilize it in time series generation in machine learning.

**Weaknesses:**

1. In experimental design, the baseline models are not sufficiently explored. The experiments should  include diffusion models for time series generation, as they are commonly used generative models nowadays as a very special variant of SDE and would provide a useful comparison to the proposed approach.

2. The space complexity and time complexity seem to raise the concerns, as signature transform is applied where the dimensionality is largely increased.

3. The scale of the datasets in the experiments are of relatively low dimensionality in time series dat. It would be hard to prove the efficiency of the adjoint method applied to reduce the memory cost.

**Questions:**

1. How should the "order" of the signature, k, be selected? Is there any specially designed optimization method to help with it?

2. Why are the neural networks in neural SDE the regular neural networks? I am very curious if more sophisticated deep learning models, such as u-nets, are examined and what their impacts to the performance are.

**Limitations:**

The limitations of this work is not adequately addressed. It can be possible to consider the application scope of this work, where it is only for time series generation, especially for the cases where the dimensionality is not high.

---

> ### Author Rebuttal · Authors · 2023-08-09
>
> **Baselines**
>
> Please see the answer to Reviewer xQxq under the **Significance** and **Empirical evaluation** headers.
>
> **Complexity**
>
> As explained in Section 3.3, our signature kernel scores are evaluated by solving scalar-valued PDEs. Therefore, our approach circumvents the computational challenges associated with calculating high-dimensional truncated signatures and can even be applied to infinite-dimensional cases like our LOB experiments. Consequently, there is no need to select a truncation level. Please also see the response to reviewer xQXq (under the **Significance** header) where we also summarize the other benefits of utilizing signature kernel scores.
>
> **Dimensionality**
>
> While we agree that our Neural SDE model is trained on relatively low-dimensional time series data, we also include an experiment on spatiotemporal LOB processes which are infinite-dimensional. As the dimension escalates and the memory cost becomes prohibitive, we can leverage the adjoint method for training the Neural SDE model.
>
> **Architecture**
>
> We chose to keep the vector fields governing the Neural SDEs as simple MLPs, aligning with the classical architecture from [1]. We appreciate your insight and note that studying different, more sophisticated deep learning models to parametrize the vector fields associated with a given Neural SDE is an interesting research topic.
>
> [1] Kidger, Patrick, James Foster, Xuechen Li, and Terry J. Lyons. "Neural sdes as infinite-dimensional gans." In International conference on machine learning, pp. 5453-5463. PMLR, 2021.

---

> > ### Comment · Reviewer_kdBJ · 2023-08-13
> > **On the diffusion models as baselines**
> >
> > Thank you for your response. It generally answers all my questions, except one point as follows.
> >
> > In terms of the baselines, your rebuttal to Reviewer xQxq mentioned that diffusion models suffer from the limitations on path generation with invariance to resolutions. The reason lies in the score-based nature of diffusion models. In specific, these models do not possess the canonical Lebesgue measure in infinite dimensions and hence do not have a coherent notion of density.
> >
> > Would you mind further explaining this reason with more details and less assumption of the audience's mathematical background knowledge? Thanks a lot!

---

> > > ### Author Response · Authors · 2023-08-13
> > >
> > > Certainly! The limitations of score-based continuous-time diffusion models to generate data in a resolution-invariant way are precisely what our Neural SDE model addresses, so it is important to clarify this point. For the sake of brevity, we will reference [1] to avoid replicating the mathematical expressions in this response. The crux of training diffusion models using scores hinges on a result from Anderson [2] stating that the inversion of a diffusion process is still a diffusion process — only running backward in time. This process's drift can be described through the gradient of the density of the marginals, $\nabla_x \log p_t(x)$ (i.e. the "score"), as highlighted in equation (6) of [1]. However, when the state X evolves in an infinite-dimensional space (like paths in our context), the notion of density $p_t$ becomes ambiguous due to the absence of a canonical Lebesgue measure. One could, in theory, adopt an alternative reference measure on pathspace, such as the Wiener measure for instance, and derive the Radon-Nikodym derivative (if it exists) against this measure. Yet, such exploration remains uncharted territory, likely necessitating a deep mathematical dive. We hope this response answers your question.
> > >
> > > **References**:
> > >
> > > [1] Song, Yang, et al. "Score-based generative modeling through stochastic differential equations." arXiv preprint arXiv:2011.13456 (2020).
> > >
> > > [2] Brian D O Anderson. "Reverse-time diffusion equation models." Stochastic Process. Appl., 12(3): 313–326, May 1982.

---

> > > > ### Comment · Reviewer_kdBJ · 2023-08-21
> > > >
> > > > Thanks for answering my question. I would like to raise my score by 1.

---

### Official Review · Reviewer_obEQ · 2023-07-24

**Soundness:** 3 good
**Presentation:** 3 good
**Contribution:** 4 excellent
**Rating:** 7
**Confidence:** 1

**Summary:**

This paper introduces a novel approach for training Neural SDEs (Stochastic Differential Equations) as generative models for sequential data without using adversarial techniques. The authors propose a novel class of scoring rules based on signature kernels, which offer stability and avoid issues like mode collapse that are common in GAN-based training. The new formulation allows for memory-efficient adjoint-based back-propagation and enables the generation of spatiotemporal data, demonstrating superior performance over alternative methods in various tasks, including simulation of rough volatility models, conditional probabilistic forecasts of forex pairs, and mesh-free generation of limit order book dynamics.

**Strengths:**

- The authors propose a new method with rigorous theoretical justification and verify it with numerical experiments.

- The proposed method does not require any adversarial training, so presumably training the generative model with the method is more stable than GANs.

- The proposed method achieves significantly better performance than SDE-GAN without adversarial training.



**Weaknesses:**

I am not expert of this field, stochastic differential equations (SDE) for continuous-time generative models. So it is difficult to point out some critical weakness of the proposed method.

-  I highly recommend that authors elaborate the evaluation metrics (e.g. KS-score) in detail for those who are not familiar with this field.
`



**Questions:**

- Is the proposed method applicable to generating images?

- How is the proposed model sensitive to hyper-parameters and what would be rule of thumbs to tune those hyperparameters?

**Limitations:**

Since my expertise is not stochastic differential equation, it is hard for me to point out critical limitations.

---

> ### Author Rebuttal · Authors · 2023-08-09
>
> **Evaluation Metric**
>
> The two-sample Kolmogorov-Smirnov (KS) test is a nonparametric statistical test used to determine whether two sets of samples come from the same continuous distribution on $\mathbb{R}$. The KS test statistic is the maximum absolute difference between two empirical cumulative distribution functions (CDFs). We appreciate that every reader may not be familiar with the KS test and are happy to amend the paper to include this explanation in the Appendix.
>
> **Image Generation**
>
> Yes, our method can be applied to images sampled at an arbitrary spatial resolution. As a training objective, one would use the scoring rule associated with one of the three kernels in $L^2$ used in the LOB example, and consider the solution of the Neural SPDE only at the final time instead of the whole solution trajectory.
>
> **Hyperparameters**
>
> In general, one can bucket the hyperparameters of the network into two categories: the “nice to have more” type (width and depth of neural networks governing the vector fields of the Neural SDEs, steps to train the model, dyadic order of the PDE solver associated to the signature kernel, step size in the generator SDE solver) which are largely constrained by computational considerations.
> The other category is the set of hyperparameters which require more careful calibration. These include which path transformations to apply, and the appropriate learning rate to select. As with any generative model, successful training is especially sensitive to these choices. Due to space constraints, we did not go into this much detail in the body of the paper. However, in the preamble to Appendix B (and in Appendix B.1-5) we provide some general rules of thumb to choose these hyperparameters, including explicit choices to replicate the results we obtained. We also provide relevant commentary in each of the notebooks found in the code repository accompanying the paper.

---

> > ### Comment · Reviewer_obEQ · 2023-08-18
> > **Response to the rebuttal**
> >
> > Thank you for your answers. I keep my current score but with low confidence score as before.

---

### Official Review · Reviewer_xQXq · 2023-08-01

**Soundness:** 3 good
**Presentation:** 2 fair
**Contribution:** 2 fair
**Rating:** 5
**Confidence:** 3

**Summary:**

This paper considered the training of neural SDEs, where are continuous-time generative models for sequential data. State-of-the-art approaches train neural SDEs in an adversarial manner and suffer from instability. In this work, a non-adversarial training approach is proposed for stable and effective training of neural SDEs. Particularly, a new class of scoring rules based on signature kernels is defined and used as objective for training. Extensive evaluations are performed to demonstrate the effectiveness of the proposed training objective.

**Strengths:**

The differences between the related works and the proposed method are clearly discussed.

The proposed method seems solid in theory.

The possible extensions of the proposed method are extensively discussed.


**Weaknesses:**

The significance of the work is somehow unclear to me. In the related work section, two works [PADD21] and [BO21] are discussed that are closely related to the proposed method. They are different from the proposed method in terms of the data type (discrete/continuous) and the setting of continuous-time processes (general/neural SDE specific). However, it is not clear how significant the proposed method is from the above two aspects.

The related work section is hard to follow. Lots of works are discussed from different perspectives: Neural SDE; score-based generative models; and scoring rules for generative networks. For me, these discussions are a bit disjointed. They are mainly discussed to show the differences between them and the proposed method. I think it is necessary to clearly locate the proposed method in the related literatures and then discuss. If there is a table or figure to visualize the overlaps between the proposed method and the other related methods, it will be clearer.

The proposed method is heavy in math. The notations are not clearly explained. For example, in line 95, what is Omega, F, and P? What is the law representing in line 97? More intuitive descriptions and clear definitions will greatly help the reading.

Though the mathematical foundation is solid, the empirical evaluation of the proposed method compared to SOTA methods is weak. Only SDE-GAN is considered for comparison. Score-based generative models (SGMs) are also leveraging scoring rules. Also, [BO21] introduced scoring rules for continuous-time processes. Since these works are closely related to the proposed method, it is necessary to compare with them. If can’t, it is necessary to explain the reasons, which I think also help clarify the significance of the proposed method.

There is a lack of experimental analysis. In section 4.1., there is no explanation or analysis for the values shown in Table 1. Why SDE-GAN is better at t=19? Similarly, in section 4.4, the conclusions of the evaluation discussed in section 4.4 are unclear to me.

The authors are claiming that the proposed method requires less memory compared with SOTAs. There is a lack of evaluation in terms of computational cost or the memory cost to support this claim.


**Questions:**

Please see the details questions in the weakness section. I would love to change my rating if the authors can address my concerns about the significance and the evaluation.

**Limitations:**

The limitations of the proposed method are well discussed.

---

> ### Author Rebuttal · Authors · 2023-08-09
>
> **Significance**
>
> We believe the innovation of our work is twofold:
>
> 1) the introduction of a new class of scoring rules for infinite-dimensional spaces of paths using signature kernels, adaptable to spatiotemporal signals, and with strict properness and consistency guarantees;
>
> 2) the deployment of these scores to train Neural SDEs, resulting in a novel generator-discriminator pair which is mesh-free, offers memory-efficient backpropagation and surpasses other Neural SDE training methods in terms of stability and performance.
>
> As mentioned in lines 80-84, in [PADD21] the authors construct statistical scores for discrete sequences, with strict properness guarantees only ensured under stringent Markov-type assumptions. A key aspect of our work is to develop consistent and proper scoring rules in the continuous-time, non-Markovian setting and use these in the context of generative modelling for functional data. Meanwhile, [B021] investigates scoring rules for continuous-time processes using truncated signatures, not signature kernels. Due to the truncation $N$ of the signature, these scores aren't strictly proper. Moreover, approaches based on truncated signatures are hindered by processes with values in $d$-dimensional spaces where $d$ is even moderate, because of the signature's exponential complexity $\mathcal{O}(d^N)$. Approaches based on truncated signatures are unusable for infinite-dimensional cases like our LOB experiments. In contrast, our signature kernel scores sidestep these computational challenges.
>
> **Related work**
>
> Our work clearly positions itself at the nexus of Neural SDEs (classically trained as GANs), and continuous-time diffusion models (classically trained using scoring rules). It's important to note that, while we reference continuous-time diffusion models, they inherently possess limitations when it comes to generating paths in a resolution invariant manner as we do. This limitation arises from the absence of a canonical Lebesgue measure in infinite dimensions which precludes a coherent notion of density (i.e. classically defined, in finite-dimensional measure theory, as the Radon–Nikodym derivative with respect to the Lebesgue measure), which is fundamental to the foundation of all score-based diffusion models.
>
> **Notation**
>
> While we respect and value the perspective of each reviewer, we note that there are differing opinions on this matter.
>
> - Reviewer YPT1 states that *The theoretical part of the paper is well-written, with clear mathematical formulations and adequate definition of symbols*.
> - Reviewer kdBJ says that *This paper is written in high clarity. All formulations and derivations are clearly elaborated.*
>
> To answer specifically the questions raised:
>
> - Line 95, as clearly stated $(\Omega, \mathcal{F}, \mathbb{P})$ is a probability space.
> - Line 97, classical definition of law of a process.
>
> **Empirical evaluation**
>
> As explained above, score-based diffusion models do not apply to generations of functional data such as paths. Besides, the scoring rules introduced in [BO21] are based on truncated signatures, which is precisely the second baseline we consider in the experiments.
>
> **Experimental analysis**
>
> Due to space constraints, we were not able to present the majority of our experimental analysis in the body of the paper. However, we do provide a more comprehensive analysis in each of the relevant sections of Appendix B, including a discussion of the hyperparameters used to achieve each set of results, further evaluation metrics, and discussion regarding both the qualitative and quantitative results we obtained. We also provided comprehensive Jupyter notebooks corresponding to each of the examples presented in the body (unconditional, conditional, and spatiotemporal generation). Remarks regarding the results found in Table 1 can be found in the paragraph directly above it. We appreciate that the reader may not be familiar with the Kolmogorov-Smirnov test; we will include an explanation for it in the final version of paper. Regarding why SDE-GAN performs better at the t=19 marginal: although SDE-GAN indeed performs better, our model also performs close to the test threshold acceptance level of 5% at the same marginal, indicating that outperformance is likely immaterial. SDE-GAN performs significantly worse at later marginals, as shown in Appendix B.1. Regarding the concern related to section 4.4 we refer to our answer to Reviewer YPT1.
>
> **Memory**
>
> We note that we do not claim anywhere in the paper that our method requires less memory than SDE-GAN, but rather that it belongs to the class of Neural SDEs, which offer memory-efficient backpropagation (aka *adjoint methods* or *optimise-then-discretise*). We also note that the computational cost of the signature-PDE solver associated to our discriminator can be found in [1] and a general discussion of the computational complexity associated to solving Neural SDEs can be found in [2]. We will include these considerations in the final version of the paper.
>
> **References**
> [1] Salvi, Cristopher, Thomas Cass, James Foster, Terry Lyons, and Weixin Yang. "The Signature Kernel is the solution of a Goursat PDE." SIAM Journal on Mathematics of Data Science 3, no. 3 (2021): 873-899.
>
> [2] Kidger, Patrick. "On neural differential equations." arXiv preprint arXiv:2202.02435 (2022).

---

> > ### Comment · Reviewer_xQXq · 2023-08-16
> >
> > Thanks for the authors’ responses. My questions are all answered. However, I still have a concern related to the presentation of the paper. For an audience who may not be deeply familiar with the concepts presented, I find the current presentation is not easy to follow. In my opinion, it is necessary for the authors to improve the presentation by minimizing assumptions about the readers’ background knowledge.
> >
> > Overall, I think this work is novel and theoretically solid. Given the authors’ responses and potential improvements to the presentation, I would like to raise my score.

---

### Official Review · Reviewer_YPT1 · 2023-08-01

**Soundness:** 3 good
**Presentation:** 3 good
**Contribution:** 3 good
**Rating:** 6
**Confidence:** 3

**Summary:**

This paper introduces a novel approach to training Neural SDEs using non-adversarial methods based on signature kernel scores. The authors demonstrate that the signature kernel score is strictly proper and provide consistent estimators for such scores. The effectiveness of their approach is demonstrated in various tasks.

**Strengths:**

1. The method is novel as it offers a non-adversarial alternative to traditional GAN-based training methods for Neural SDEs. This work is likely the first to introduce the signature kernel method to the Neural-SDE model. It achieves superior performance compared to the GAN method while being easier to train.

2. The theoretical part of the paper is well-written, with clear mathematical formulations and adequate definition of symbols. The proof of strict properness and consistency, although direct consequences of prior works, offers robust guarantees.


**Weaknesses:**

1. Limited comparisons: The authors claim that their procedure outperforms alternative ways of training Neural SDEs, but they only compare it with the SDE-GAN. In the related works, the authors mention the latent SDE, which is easier to train than SDE-GAN, and claim that it yields worse performance than the SDE-GAN. While I acknowledge that GAN generally has better model capacity than VAE, can you provide quantitative results to substantiate this claim? The original paper only compares with the latent ODE from 2019.

2. Lack of visualizations: Although the paper visualizes the generated distribution of paths conditioned on a previous path using their model, it would be beneficial to include a comparison with the paths generated by the SDE-GAN. This would provide a clearer understanding of how your outputs capture more characteristics of financial markets. Furthermore, the generated paths exhibit a large variance, which could be addressed or explained.

3. Explanation of the LOB experiments: The simulation of LOB dynamics and the decision not to use autoregressive prediction are interesting, but how is the proposed design better than the five autoregressive generators that are listed? The paragraph discussing this topic appears to be hastily written. The reported average scores alone do not provide sufficient information.


**Questions:**

1. Can you provide quantitative results to show latent-SDE performs worse than SDE-GAN and your model?
2. Can you visualize conditional path distributions generated by SDE-GAN and explain why your generated paths are better?
3. Could you provide more details on how your model is useful in limit order book (LOB) prediction and elaborate on the design choices you made?

**Limitations:**

The authors have well discussed the limitations. The work assumes the noise is continuous without jump on event arrival, therefore is not applicable to noise governed by point processes. The work pose certain regularization conditions on the sampled paths. The work could benefit from including more varied evaluation metrics to assess the performance of the generated data.

---

> ### Author Rebuttal · Authors · 2023-08-09
>
> **Comparisons**
>
> The comparison between SDE-GANs and latent SDEs (and the limitations of the latter) has been discussed in the PhD thesis [1, Section 4.3.3] and further analysis as well as quantitative results can be found in [2]. Therefore we decided to only include the most expressive model among all Neural SDEs in the benchmark and showcase that our approach, while non-adversarial, outperforms SDE-GANs, contrarily to latent SDEs. We will refer the reader to [1,2] to support the claim made in the related work section.
>
> **Visualizations**
>
> We did not visualize conditional path distributions generated by SDE-GAN, as this would require significant modifications to the existing SDE-GAN implementation (available in the torchsde package). The authors of SDE-GAN do present a conditional example in their work, however, it is a class-conditioning example where the conditioning variable is discrete. For conditioning data arising from a continuous variable in an infinite-dimensional space (such as paths), this procedure cannot be implemented.
>
> In the unconditional setting, our generated paths do not have a larger variance than those exhibited by the data measure (as confirmed by the Kolmogorov-Smirniov test on the marginals). This holds true in the conditional setting. We agree with your suggestion to further comment on Figure 2. Therefore, we propose to add the following remark: Although the three conditional distributions exhibit a larger variance (from left to right), this is to be expected, since they correspond to three different (increasing) levels of quadratic variation exhibited by the conditioning paths.
>
> **LOB Experiment**
>
> The primary objective of the LOB experiment is to demonstrate that the newly proposed signature kernel scores can be used to effectively train continuous space-time generative models. However, we agree with the reviewer that our modelling choices can be further motivated and propose to add the following comments.
>
> Autoregressive generators produce “one-step-ahead” predictions by parameterizing the conditional distribution of a process at a specific point in time, conditioned on the previous k observations. Instead, we propose to use the Neural SPDE model as a generator which directly produces continuous time LOB trajectories in a discretization-free manner (also see the answer to Reviewer xQXq under the **Significance** header). Furthermore, we note that recent works in mathematical finance [3,4] have established that LOB dynamics are well-described by mechanistic models in the form of SPDEs. In light of these results, the Neural SPDE model is a natural choice for the generator as it offers a strong prior on the model space.
>
> We report the KS test statistic for each individual marginal, both in the spatial and temporal dimensions, in line with the other experiments in the paper. We recognize that it would be beneficial to include additional evaluation metrics. However, well-established and general-purpose metrics (that do not require expert knowledge on the specific task at hand) for assessing functional spatiotemporal generative models are still lacking. This gap arises due to the novelty of this field. Potential approaches could involve computing spatiotemporal semivariograms. However, we did not find adequate Python libraries for our purpose. Another direction to explore as future work would be to employ a “train on synthetic, test on real” methodology [5]. We will mention this in the future work section.
>
> **References**
>
> [1] Kidger, Patrick. "On neural differential equations." arXiv preprint arXiv:2202.02435 (2022).
>
> [2] Kidger, Patrick, et al. "Efficient and accurate gradients for neural sdes." Advances in Neural Information Processing Systems 34 (2021): 18747-18761.
>
> [3] Hambly, Ben, Jasdeep Kalsi, and James Newbury. "Limit order books, diffusion approximations and reflected SPDEs: from microscopic to macroscopic models." Applied Mathematical Finance 27.1-2 (2020): 132-170.
>
> [4] Cont, Rama, and Marvin S. Müller. "A stochastic partial differential equation model for limit order book dynamics." SIAM Journal on Financial Mathematics 12.2 (2021): 744-787.
>
> [5] Esteban, Cristóbal, Stephanie L. Hyland, and Gunnar Rätsch. "Real-valued (medical) time series generation with recurrent conditional gans." arXiv preprint arXiv:1706.02633 (2017).

---

### Author Rebuttal · Authors · 2023-08-09

We thank all the reviewers the useful feedback and insightful questions. We hope that our responses will address all raised concerns.

---

### Decision · Program_Chairs · 2023-09-21

**Decision:**

Accept (poster)

**Comment:**

The paper proposes a non-adversarial training strategy for neural SDEs. The reviewers think the writing is clear and the method based on signature Kernel scores is novel. Please elaborate the evaluation metrics and provide more background introduction in the final version.